# The Representation of Rivers in Operational Ocean Forecasting Systems: A Review

Pascal Matte[1], John Wilkin[2], Joanna Staneva[3]

[1]Meteorological Research Division, Environment and Climate Change Canada, Québec, QC, Canada
[2]Department of Marine and Coastal Sciences, Rutgers, The State University of New Jersey, New Brunswick, NJ, USA
[3]Institute of Coastal Systems - Analysis and Modeling, Helmholtz-Zentrum Hereon, Geesthacht, Germany

*Correspondence to*: Pascal Matte (pascal.matte@ec.gc.ca)

**Abstract.** The connection between the ocean and the land is made possible thanks to rivers, which are a vital component of the Earth's system. They govern the hydrological and biogeochemical contributions to the coastal ocean through surface and subsurface water discharge and influence local circulation and the distribution of water masses, modulating processes such as upwelling and mixing. This paper provides an overview of recent approaches to representing coastal river discharges and processes in operational ocean forecasting systems (OOFS), with a particular focus on estuaries. The methods discussed include those currently adopted in coarse-resolution ocean forecasting systems, where mixing processes are primarily parameterized, as well as more advanced modelling and coupling approaches tailored to high-resolution coastal systems. A review of river data availability is also presented, illustrating various sources of freshwater discharge and salinity, from observational data to climatological datasets, alongside operational river discharge products that enhance the representation of water discharges in operational systems. New satellite-derived datasets and emerging river modelling techniques are also introduced. In addition, responses from a survey of existing OOFS providers are synthetized, with a focus on how river forcing is treated, from global to coastal scales. Challenges such as data accuracy, standardization, and model coupling are discussed, highlighting the need for improved interfaces between monitoring and modelling systems. Finally, some recommendations and ways forward are formulated in relation to identified limitations in current OOFS.

**1 Introduction**

Rivers form the primary link between land and sea, delivering approximately 36,000 km³ of freshwater and over 20 billion tons of solid and dissolved material to the global ocean each year (Milliman and Farnsworth, 2011). River discharge into the ocean is a major component of the global hydrological and biogeochemical cycles, which have undergone significant changes under the influence of climate and human activities (Shi et al., 2019; Yan et al., 2022; Qin et al., 2022; Chandanpurkar et al., 2022). Estuaries act as transitional zones where freshwater fluxes influence ocean circulation, salinity, and upper-ocean stratification, which in turn affects the mixed layer depth, ocean currents, and air-sea interaction (Chandanpurkar et al., 2022; Dzwonkowski et al., 2017; Sprintall and Tomczak, 1992; Sun et al., 2017; Pein et al., 2021, 2024). Freshwater inputs to the

ocean also modulate coastal upwelling events. Altogether, these factors impact productivity of the coastal marine environment (Sotillo et al., 2021a).

Despite rivers' influence on the coastal and basin-wide circulation and dynamics, in global and regional scale models, effectively accounting for riverine freshwater discharge into the oceans is a challenging problem (Sun et al., 2017; Verri et al., 2020). Accurately incorporating river flow into numerical ocean models requires appropriate parameterizations and boundary conditions. The setup of practical open boundary conditions (OBC) is dependent on flow dynamics, model resolution, data availability, and other factors (Blayo and Debreu, 2005). At coarse scales that cannot resolve the estuarine dynamics, but even

at finer scales in some cases, river outlets are often represented in a simplistic way, with climatological runoff and zero or constant salinity values, implicitly neglecting estuarine mixing and exchange as well as seasonal and non-seasonal variability (Sun et al., 2017; Verri et al., 2020; Verri et al., 2021; Pein et al., 2021, 2024). As a result, key natural processes are often omitted, and depending on how river forcing is defined, ocean model outputs may vary significantly. These discrepancies are most pronounced in shelf areas, particularly in Regions of Freshwater Influence (ROFI), but can also propagate to regional

and global scales (Tseng et al., 2016).

This paper reviews existing methods and datasets used in Operational Ocean Forecasting Systems (OOFS) to represent river forcing. As the focus is on freshwater discharges, the river supply of nutrients and other materials are neglected in this review but are partly addressed in a separate contribution by Cossarini et al. (2024).

The paper is structured as follows: Section 2 reviews approaches for representing river forcing in global, regional, and coastal

ocean models, including estuarine mixing parameterizations and coupling techniques. Section 3 describes available data sources from operational centers and data providers as well as emerging techniques for estimating river discharge. Section 4 presents examples of river forcing methods and data sources implemented in existing OOFS, summarizing findings from a survey conducted within the OceanPredict community. Finally, Section 5 provides a summary and recommendations regarding identified limitations in current OOFS.

**2 River forcing in ocean models**

**2.1 Capturing seasonal and non-seasonal river variability**

Accurate representation of river discharges and associated variables (e.g., salinity, temperature), whether model-derived or observation-based, is crucial for capturing both seasonal and non-seasonal effects in the coastal ocean. The Bay of Bengal is one example where the inclusion of seasonal river discharges and salinity in regional model simulations significantly improves

the representation of sea surface temperatures, near-surface salinity, stratification, mixed-layer depth, and barrier-layer thickness, leading to a better simulation of the formation, progression and dispersion of the freshwater plume (Jana et al., 2015).

Seasonal variability in river discharge not only impacts coastal salinity and temperature but also contributes to the sea level changes both locally and remotely, mostly via a halosteric sea level contribution. This effect was observed, for example,

between the mouth of the Amazon River and the continental shelves of the Gulf of Mexico and Caribbean Sea (Giffard et al., 2019). Similarly, in the U.S. Atlantic and Gulf coasts, river discharge and sea level changes were found to be significantly correlated (Piecuch et al. 2018). Such dynamic sea surface height (SSH) signals driven by river discharge can explain 10-20% of the regional-scale seasonal variance around major rivers, such as the Amazon, Ganges, Brahmaputra, Irrawaddy, Ob, Lena, and Yenisei (Piecuch and Wadehra, 2020).

While the seasonal effects of river discharge on ocean processes have been extensively documented, non-seasonal influences of river runoff on sea level changes remain largely unexplored due to the lack of consolidated discharge databases (Durand et al., 2019). These influences, however, can be significant when considering river runoff jointly with wind-driven transport and heat fluxes, which also play a major role in modulating regional sea level variability (Verri et al., 2018).

## 2.2 Freshwater input in coarse resolution models: towards a parameterization of estuarine mixing processes

Because many ocean models operate at resolutions too coarse to resolve estuarine processes explicitly, appropriate parameterization of estuarine mixing is required to capture their influence on freshwater transport. In nature, estuaries transport and transform water properties along their length, due to tidal mixing, deposition and resuspension, and up- and down-estuary advection. Saltwater intrusion driven by tides and other coastal signals (e.g. storm surges) controls the estuarine water exchange and affects the net estuarine outflow and corresponding salinity values (Sun et al., 2017; Verri et al., 2020). However, although

water properties at the head differ from those at the mouth, in models too coarse to resolve the estuaries, river discharge observed far from the river outlet is typically inputted at the coast with zero salinity (Verri et al., 2021; Herzfeld, 2015). Alternatively, salinity values can be prescribed based on constant annual or monthly values derived from sensitivity tests and/or in situ campaigns, when available (Verri et al., 2018).

Herzfeld (2015) describes and assesses the performance of various methods for inputting freshwater into regional ocean

models. A first approach, referred to as a point source input, adds a term of freshwater flux, entering as surface point sources into one or more layers of the model, to the divergence of flow in the vertically integrated continuity equation, with no associated velocity profile. It affects the vertical velocity surface boundary condition of the free surface equation, and the surface boundary conditions for the diffusive heat and salt fluxes. A second approach, the flow input, considers the inertia of the river flow and prescribes a velocity profile at the boundary whose vertical integral is equal to the inflow flux. These two

methods must have a predefined depth at the boundary over which to distribute the volume inflow. A more accurate approach is to add an artificial channel to the coastline to give momentum to the flow and initiate mixing between fresh and salt waters (Lacroix et al., 2004; Sobrinho et al., 2021).

The horizontal distribution of the runoff plays an important role in the regional salinity distribution and in the vertical stratification and mixing (Tseng et al., 2016). Additional subtleties arise for large rivers or deltas, where the coastal source

points need to be spread laterally to avoid numerical instabilities if inflow values are locally too large (Polton et al. 2023). In global ocean models, however, freshwater inflow is frequently added at the ocean surface, either as an increased precipitation rate over a specified area or by reducing surface salinity (i.e. a virtual salt flux), rather than being introduced as a lateral inflow

at the coastal boundary. This freshwater can be distributed vertically over several layers or diffused horizontally using enhanced mixing (Sun et al., 2017; Tseng et al., 2016; Yin et al., 2010).

Several plume responses may result from the choice of the horizontal and vertical distribution of freshwater input. However, most model applications produce plumes whose types differ from plumes associated with real river discharges (Tseng et al., 2016; Garvine, 2001; Schiller and Kourafalou, 2010). Larger scale offshore stratification is also expected to be impacted by this choice.

MacCready and Geyer (2010) establish the theoretical foundation for estuarine mixing parameterizations, which underpins
some physics-based methods used to simulate unresolved estuarine processes in regional and global ocean models, such as the estuary box model (EBM); see, for example, Figure 1 (Sun et al., 2017). These models attempt to parameterize mixing processes and to account for baroclinic and barotropic flow, typically using a two-layer formulation (e.g. Verri et al., 2020; Verri et al., 2021; Herzfeld, 2015; Rice et al., 2008; Hordoir et al., 2008). From these representations, analytical solutions can be found for the volume fluxes and outflow salinity. Applied globally to the Community Earth System Model (CESM), such
an approach revealed substantial localized, regional, and long-range effects when compared to cases without parameterization, highlighting once again the strong sensitivity of ocean models to the treatment of rivers (Sun et al., 2017).

New hybrid approaches, such as the Hybrid-EBM (Maglietta et al., 2025; Saccotelli et al., 2024), combine physics-based models with machine learning techniques to predict the salt-wedge intrusion length and salinity at river mouths. Hybrid-EBM outperforms the classical EBM and addresses the shortcomings of the dimensional equations in the physics-based EBM, which
rely on several tunable coefficients and require site-specific calibration, by substituting them with machine learning algorithms (Maglietta et al., 2025).

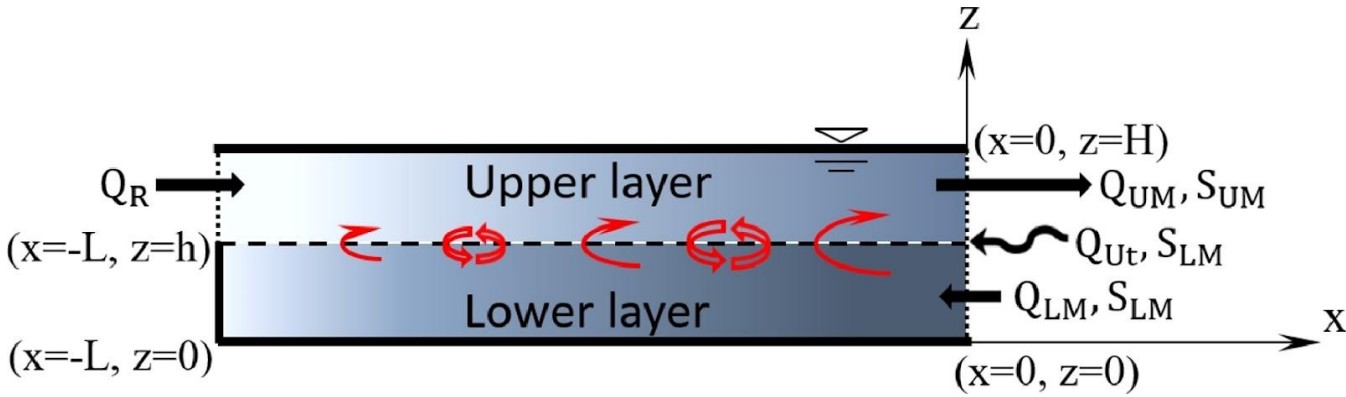

**Figure 1: Schematic diagram of the estuary box model (EBM) implemented in the Community Earth System Model (CESM) (Sun et al., 2017). The EBM is depicted as a two-layer rectangular box with constant width, uniform local depth (H), and a time-varying**
**length (L). Each layer has a fixed thickness (h for the lower layer and H-h for the upper layer), with vertically uniform but horizontally variable salinity and density. Thick solid lines represent closed boundaries, dotted lines mark open boundaries, and the dashed line shows the interface between layers. Volume fluxes (Q) and salinities (S) are indicated by arrows at open boundaries: riverine freshwater discharge (Q$_R$) enters at the estuary head, oceanic saltwater flows into the lower layer at the mouth (Q$_{LM}$), and Q$_{Ut}$ represents the average tidal volume flux during half a tidal cycle, driving net horizontal salt flux into the upper layer at the**
**mouth. Shear-induced turbulent mixing (shown by paired upward and downward open arrows) and upward advection from**

**exchange flow (solid upward arrows) link the upper and lower layers. The color gradient illustrates salinity variation, from fresher (lighter shades) to saltier (darker shades) waters.[1]**

### 2.3 Freshwater input in high resolution models: unstructured modelling of the river-sea continuum

In contrast, when the model resolution is higher than the estuary width, the latter can be resolved explicitly by extending the
grid for some distance inland using either real bathymetry or a straight channel approximation. When extending it beyond the salinity intrusion limit and/or the head of tides, a freshwater flux can be directly specified at the upstream boundary. This is the preferred option in many east coast US studies (Herzfeld, 2015) (e.g. RISE - Liu et al., 2004; LATTE - Choi and Wilkin, 2007; MerMADE - Hetland and MacDonald, 2008).

The use of unstructured grids offers various advantages, including a more accurate treatment of the freshwater inputs from
rivers, a realistic representation of river-sea interactions and estuarine processes at spatial and temporal scales usually not resolved in the ocean, and an improved interface between estuaries and the open ocean, sometimes with higher-order spatial discretizations (Staneva et al., 2024). In addition, the unstructured grid modelling combined with an efficient vertical coordinate system can better resolve the coastal sea dynamics (Verri et al., 2023).

With seamless grid transitions between models or domains, flexibility and cross-scale capabilities are augmented (Zhang et
al., 2016). As examples, a river-coastal-ocean continuum model has been developed for the Tiber River delta, reproducing the coastal dynamic processes better than the classic coastal–ocean representation, including the salt wedge intrusion, and revealing new features near the river mouth induced by river discharge and coastal morphology (Bonamano et al., 2024). In the Columbia River estuary, where both shelf and estuarine circulations are coupled, a multi-scale model has proved to reproduce key processes driving the river plume dynamics in a region characterized by complex bathymetry and marked
gradients in density and velocity (Vallaeys et al., 2018). Likewise, Vallaeys et al. (2021) used a similar model in a topographically challenging area of the Congo River estuary, characterized by high river discharge, strong stratification and large depth. Similarly, Maicu et al. (2021) simulated the circulation in the Goro Lagoon and Po River Delta branches using downscaling and a seamless chain of models integrating local forcings and dynamics into a coarser OOFS based on a cascading approach.

While these examples were successful in representing dynamical processes across temporal and spatial scales, in some contexts, the large inward tidal extent and/or complex bathymetries and coastlines, often featuring coastal infrastructures, pose significant challenges for explicitly resolving estuaries, making it impractical in many coastal models. As a result, this approach has yet to become standard practice in OOFS.

---

[1] Reprinted from Ocean Modelling, Vol 112, Sun, Q., Whitney, M. M., Bryan, F. O., and Tseng, Y., A box model for representing estuarine physical processes in Earth system models, Page 140, Copyright Elsevier Ltd. (2017), with permission from Elsevier.

**2.4 One-way and two-way coupling**

Coupling techniques can be used to link two or more models to allow one-way data exchange, for example, between a hydrological model and an ocean model. In this approach, external forcing is reduced to a limited set of variables, simplifying computational requirements but potentially overlooking key processes at the land-sea interface. Additionally, it requiresextending the ocean domain boundaries far inland, beyond the limit of tide and storm-surge propagation. While some parameterizations (cf. Section 2.2) or use of unstructured grids (cf. Section 2.3) can partly alleviate these shortcomings, in a

compound flooding context, two-way coupled models are preferred because both land and ocean processes can be represented along with their interactions (Bao et al., 2022; Cheng et al., 2010). The inclusion of momentum flux exchanges between land and ocean improves the simulation of estuarine water levels by capturing nonlinear feedbacks between runoff and residual ocean water levels. In a case study of Hurricane Florence, Bao et al. (2022) achieved significant improvement in simulated water levels (20%-40% at the head of Cape Fear River Estuary) during the post-hurricane period by using a two-way coupled

model, compared to a stand-alone and linked (one-way coupled) approach.

Alternative approaches for assessing the risk of compound flooding have been proposed, including integrated hydrodynamic and machine learning methods to predict water level dynamics (Sampurno et al., 2022). Such approaches are particularly valuable in data-scarce regions, where developing fully calibrated, computationally intensive models can be impractical or infeasible.

**3 Data sources**

**3.1 Freshwater discharge**

A persistent challenge in OOFS with respect to river forcing is the lack of a global network for observed river flows to the oceans. While advances are being made in creating such a network, several challenges remain pertaining to data quality, accessibility, and timeliness, at the required spatial and temporal scales.

In situ river discharge observations are necessary to build climatologies. They represent a key component of the calibration of hydrological models, and thereby of any reanalysis, near-real-time (NRT) analysis and forecast products. The various types of discharge products used in OOFS are described in the following.

**3.1.1 Climatologies**

Most ocean models use climatologies to introduce river forcing based on multi-decadal averages of observed and/or modeled

freshwater discharges, along with zero or constant salinity values. Although climatological data is commonly used, even in cases where estuarine dynamics are not explicitly resolved, more realistic volume flux and salinity estimates would improve the modelling of coastal (e.g. river plumes) to basin-wide circulation and dynamics (e.g. dense water formation, overturning circulation cells, water exchange at straits) (Verri et al., 2018), especially during non-seasonal (e.g. storm induced) events

(Chandanpurkar et al., 2022). Moreover, given the global decline of the hydrometric networks, building climatologies is not always possible, especially for small or less-studied rivers, and even for large rivers in regions where routine monitoring is absent (Campuzano et al., 2016; Mishra and Coulibaly, 2009). Furthermore, monthly climatological products are not adequate for high resolution coastal models where temporal variability at daily or even higher frequency is needed (Sotillo et al., 2021a).

### 3.1.2 River discharge databases

In contrast, river databases and services are progressively becoming available and provide better estimates of coastal runoff and river discharges at the global scale (Sotillo et al., 2021a). These databases typically assemble information from multiple data providers into coherent, gap-free and quality-controlled datasets. Examples below are categorized by data source:

*In situ databases:*

- The Global Runoff Data Center[2] (GRDC), under WMO, archives quality-controlled historical mean daily and monthly discharge data from over 10,000 stations across 159 countries. The Freshwater Fluxes into the World's Oceans[3] dataset, based on the water balance model WaterGAP, provides annual runoff estimates from 1901-2016.

- The Global Streamflow Indices and Metadata archive (GSIM), a collection of metadata and indices derived from more than 35 000 daily streamflow time series worldwide, gathered from 12 open databases (7 national and 5 international collections) (Do et al., 2018; Gudmundsson et al., 2018).

- A global dataset of monthly streamflow for 925 of the world's largest rivers connecting to the ocean was built by Dai et al. (2009), updated from Dai and Trenberth (2002).

- A global database of monthly mean runoff for 986 rivers was incorporated in the NCOM, now HYCOM, U.S. model (Barron and Smedstad, 2002). It expands on the work of Perry et al. (1996) with corrections and additions derived from monthly mean streamflow from the U.S. Geological Survey (USGS) (Wahl et al., 1995), and extends the basic RivDIS database (Vörösmarty et al., 1998) to adjust for missing discharge attributed to small (ungauged) rivers.

*Model-derived databases:*

- A 35-year daily and monthly global reconstruction of river flows (GRADES) at 2.94 million river reaches, with bias correction from machine-learning derived global runoff characteristics maps, was developed in support of the Surface Water and Ocean Topography (SWOT) satellite mission (Lin et al., 2019).

- A dataset of historical river discharge from 1958 to 2016 was created using the CaMa-Flood global river routing model and adjusted runoff from the land component of JRA-55 (Suzuki et al., 2018; Tsujino et al., 2018).

- A global freshwater budget is included in the CORE.v2 datasets that have an accompanying database for continental runoff from rivers, groundwater and icebergs. These are estimated from continental imbalances between precipitation,

---

[2] https://grdc.bafg.de/
[3] https://fwf.bafg.de/

evaporation and storage, and then distributed between bordering ocean basins based on river routing schemes and flow estimates (Large and Yeager, 2009).

*Hybrid database:*

- EMODnet Physics[4] provides ocean physics data and data products built with common standards, consisting of collections of in situ data, reanalysis, and aggregated in situ data and model outputs. As part of the available parameters, the operational river runoff data includes near-real time data from European river stations and a subset of the GRDC focusing on coastal areas and stations located near river mouths, which extend beyond European borders.

About 1,200 rivers worldwide are connected and operationally available.

*Satellite-derived database:*

- The largest known dataset compiles publicly available river gauge data, with satellite-based rating curves used to fill in the temporal gaps (Riggs et al., 2023).

Regional databases also exist, such as:

- Long-term (1993-2011) satellite-derived estimates of continental freshwater discharge into the Bay of Bengal (Papa et al., 2012).
- A database of pan-Arctic river discharge (R-Arcticnet[5]).
- A database for Greenland liquid water discharge from 1958 through 2019 (Mankoff et al., 2020).
- A river discharge climatology and corresponding historical time series for all rivers flowing into the Adriatic Sea with

an average climatological daily discharge exceeding 1 $m^3s^{-1}$ (Aragão et al., 2024).

Of particular importance is the fact that some of these databases use model-simulated runoff ratios (e.g. from Community Land Model (CLM) or river routing model) over gauged and ungauged drainage areas to estimate the contribution from the areas not monitored by the hydrometric network and adjust the station flow to represent river mouth outflow (e.g. Dai et al. 2009). This allows more precise derivation of the total discharge into the global oceans, through the sum of both gauged and ungauged

discharges.

Unless explicitly stated (e.g. for EMODnet Physics), most of these databases lack clearly stated update schedules; some remain static, while others update at irregular intervals. Such databases are useful in the context of a reanalysis, but less so in an operational context where near-real-time data feeds are required. Furthermore, a detailed comparative assessment of these various data sources is still lacking.

Alternatively, indirect approaches using tidal statistics at the estuarine entrance from tidal stations rather than direct flow measurements have been developed to estimate the net freshwater discharge at the mouth of an estuary, with the advantage of integrating processes at the basin scale, downstream of the last hydrometric station (Moftakhari et al., 2013; Moftakhari et al.,

---

[4] https://emodnet.ec.europa.eu/geoviewer/
[5] https://www.r-arcticnet.sr.unh.edu/v4.0/index.html

2016). Because tide gauge records at the coasts were often installed well before the onset of systematic river gauging (Talke and Jay, 2013), such inverse techniques make it possible to extend flow records back in time.

### 3.1.3 Operational river discharge products

While most river discharge databases are static, operational products have been developed for near-real-time applications. For example, the Global Flood Awareness System, GloFAS-ERA5, is an operational global river discharge reanalysis produced consistently with the ECMWF ERA5 atmospheric reanalysis and providing global gridded data products from 1979 to near-real-time (within a 7-day delay) (Harrigan et al., 2020). Figure 2 illustrates the resolution of the river network that emerges in the GloFAS gridded data, and the association of discharge at the coast to point sources in a regional model of the northwest Atlantic Ocean that is in development for future operations.

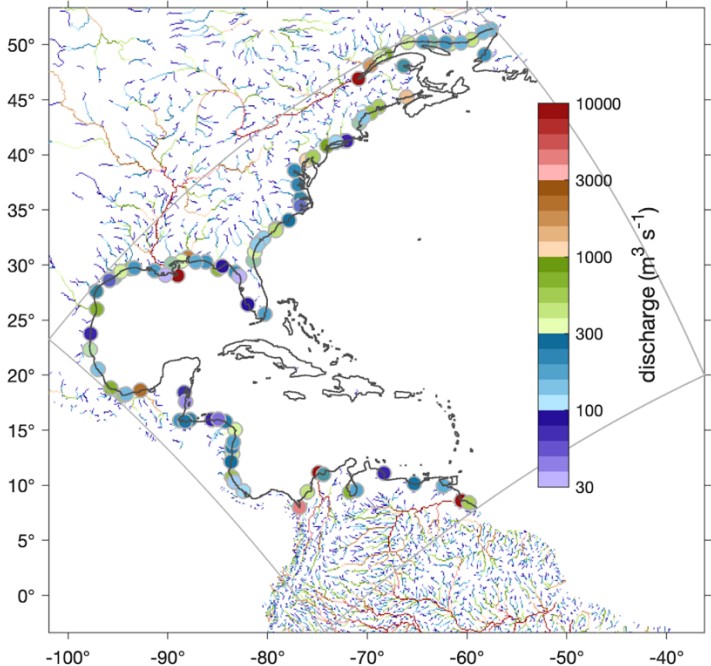

**Figure 2: Annual mean surface water discharge (m³s⁻¹) in 0.1° x 0.1° cells of the GloFAS analysis from Harrigan et al. (2020) for the year 2023. Filled circles show the locations of 93 point sources in the prototype East Coast Community Ocean Forecast System (ECCOFS) ROMS model (domain denoted by the gray perimeter box) associated to GloFAS points near the coast that have long-term mean (2009-2019) discharge exceeding 50 m³s⁻¹. River networks come from GloFAS.**

Several centers are also producing continental- and global-scale hydrological (ensemble) forecasts operationally: the European Flood Awareness System (EFAS) (Thielen et al., 2009), the European Hydrological Predictions for the Environment (E-HYPE) (Donnelly et al., 2015), the Hydrologic Ensemble Forecast Service (HEPS) in the U.S. (Demargne et al., 2014), the Flood Forecasting and Warning Service (FWWS) in Australia, the National Surface and River Prediction System (NSRPS) in Canada

(Fortin et al., 2023); and globally, the World-Wide HYPE (WWH) (Arheimer et al., 2020) and GloFAS (Harrigan et al., 2023). Notably, as part of the GloFAS service evolution, global daily ensemble river discharge reforecasts (20-year) and real-time forecast (2020-present) datasets are made freely and openly available through the Copernicus Climate Change Service (C3S) Climate Data Store (CDS) (Harrigan et al., 2023).

Other projects have been supported by the Copernicus Marine Environment Monitoring Service (CMEMS), for example, the LAMBDA project regionally focused on the European Atlantic Façade and the North Sea. The resulting freshwater model estimates and in-situ observations are operationally updated and made available via the project viewer web interface[6] (Sotillo et al., 2021a).

The FOCCUS project (Forecasting and Observing the Open-to-Coastal Ocean for Copernicus Users[7]) further enhances
operational hydrological models by addressing the land-ocean continuum through improved river runoff estimations and the development of advanced coupling between hydrological and coastal ocean models. FOCCUS builds on existing pan-European hydrological frameworks, such as E-HYPE and LISFLOOD, to provide dynamic freshwater inputs, including nutrient and inorganic matter transport. Additionally, the project integrates novel AI techniques to optimize estuarine modelling and freshwater forcing for coastal systems. These innovations directly contribute to refining CMEMS and supporting all European
coastal services with more accurate and seamless coastal monitoring and forecasting capabilities.

In some instances, the regional products may appear to be the preferred option for some regional or local studies, as they were designed to specifically represent the hydrological characteristics of a given region, sometimes with higher resolution and accuracy. However, a global solution is attractive in data scarce areas and where consistency between discharge products and across all forcing variables is required over large domains (Polton et al., 2023).

**3.1.4 Remotely-sensed discharges**

Remote sensing of river discharge is a rapidly advancing research field (see Gleason and Durand, 2020, and references therein). With the SWOT satellite launched in December 2022, global discharge products will soon be available at a nominal resolution of 10 km for river reaches wider than 100 m, thus vastly expanding measurements of global rivers in both gauged and ungauged basins (Durand et al., 2023). Significant improvements on global uncalibrated models are expected (Emery et al., 2018).
SWOT-derived discharge data is expected to improve global hydrological cycle representation and enhance ocean model solutions near the coast.

**3.1.5 Machine learning-derived discharge estimates**

Machine learning is increasingly used in hydrology for rainfall-runoff modelling, with Long Short-Term Memory (LSTM) networks (Greff et al., 2016; Hochreiter and Schmidhuber, 1997) proving particularly effective in capturing both periodic and

---

[6] http://www.cmems-lambda.eu/home.html
[7] https://foccus-project.eu/

chaotic patterns in time-series data while accurately learning long-term dependencies (Fang et al., 2017; Hu et al., 2019; Mouatadid et al., 2019). In numerous hydrological studies, LSTM has demonstrated superior performance over traditional process-based models in simulating runoff, primarily in data-rich regions (Feng et al., 2020, 2021; Frame et al., 2022; Gauch et al., 2021; Hunt et al., 2022; Konapala et al., 2020; Kratzert et al., 2019; Lees et al., 2021; Li et al., 2023; Luppichini et al., 2024; Nearing et al., 2021; Reichstein et al., 2019). However, limited efforts have explored the transferability of LSTM models

to data-scarce regions (e.g. Akpoti et al., 2024), with Ma et al., (2021) and Muhebwa et al. (2024) (and references therein) being a few such exceptions. Recently, researchers have explored the potential of LSTM models for global river discharge estimations (Rasiva Koya and Roy, 2024; Tang et al., 2023; Yang et al. 2023; Zhao et al. 2021). However, extensive validation beyond the training basins is required to fully evaluate their suitability for global-scale implementations.

## 3.2 Salinity and temperature

Estuarine mixing influences salinity distribution and its seasonal variability near river mouths (Sun et al., 2019). Models are particularly sensitive to salinity in shelf areas and ROFI zones, most often due to the diverse treatment of OOFS given to coastal and river freshwater forcing (Sotillo et al., 2021a). Therefore, to assess the impact of a chosen formulation and evaluate model performances, sea surface salinity (SSS) and temperature (SST) are typically used. The World Ocean Atlas climatology (Locarnini et al., 2013; Zweng et al., 2023) often overestimates nearshore salinity, making it unsuitable for model evaluation

in coastal regions. As an alternative, Sun et al. (2019) built on the original World Ocean Database and developed an improved salinity and temperature climatology with an enhanced representation of the coastal ocean. In-situ data and satellite observations from SMOS, Aquarius and SMAP (Bao et al., 2019) can also be used to assess the impact of river forcing on sea surface salinity (Feng et al., 2021). However, seasonal variability in the skill of SSS retrievals can be associated with SST-dependent bias and strong land-sea differences in microwave emissivity, making satellite observations unreliable within some

70 km of the coast (Grodsky et al., 2018; Menezes, 2020; Vazquez-Cuervo et al., 2018). Higher resolution coastal satellite products have been developed based on empirical relationships between local salinity and ocean color observations (Geiger et al, 2011; Chen et al., 2017), using deep neural networks trained on Sentinel-2 Level 1-C Top of Atmosphere (TOA) reflectance data (Medina-Lopez and Ureña-Fuentes, 2019; Medina-Lopez, 2020), or by relating the reflectance of the visible bands from Sentinel-2 imagery with electrical conductivity, influenced by the concentration and composition of dissolved salts (Sakai et

al., 2021), although these are not applied globally.

A recent study in the German Bight (Thao et al., 2024) demonstrated the critical role of high-resolution salinity inputs at estuarine mouths in improving the predictive capabilities of coupled wave-ocean models. Using the GCOAST model system, which seamlessly integrates estuarine and coastal dynamics with regional ocean models, researchers validated salinity and temperature fields against in-situ observations. The results highlighted that estuarine inflows significantly enhance the

accuracy of coastal ocean models.

Alternatively, salinity predictions in estuaries and at river mouths have been successfully estimated using machine learning approaches. A few examples can be found in the recent literature: Qiu and Wan (2013) developed an autoregressive model

relating salinity at a given time to past observations of salinity and physical drivers (freshwater inflow, rainfall, tidal elevation) in the Caloosahatchee River Estuary; Fang et al. (2017) used a genetic algorithm coupled with support vector machine to predict salinity in the Min River Estuary; Qi et al. (2022) applied four neural network models to emulate salinity simulations in the Sacramento-San Joaquin Delta from a process-based river, estuary and land modelling system; Guillou et al. (2023) were able to reproduce the seasonal and semi-diurnal variations of sea surface salinity at the mouth of the Elorn estuary (bay of Brest), with support vector regression performing best among all tested algorithms.

Despite these advancements, sustained high-resolution salinity monitoring is needed to build confidence in numerical solutions near the coast. Integrating salinity, temperature, and additional parameters such as nutrients and sediments directly into river outflows could further improve model accuracy (Verri et al., 2018; Thao et al., 2024). While these factors play a secondary role in influencing oceanographic processes, their inclusion could advance research on coastal hypoxia, carbon cycling, and regional weather and climate, ultimately supporting seamless predictions of land–ocean–atmosphere feedbacks in next-generation Earth system models (Feng et al. 2021).

## 4 Examples of current OOFS

This section describes how river forcing is implemented in current OOFS. The objective is to get a picture of the current landscape of approaches and data sources. While Cirano et al. (2024) provide a comprehensive overview of existing OOFS worldwide, the representation of rivers in these systems remains poorly documented and often buried in model configuration files. The list of systems presented in Appendix A is therefore not exhaustive and is limited to a compilation of comments received as part of a survey conducted among members of the OceanPredict community in May 2023. It is meant to illustrate the diversity of methods employed for treating freshwater fluxes in OOFS and associated input data sources, in 4 global, 12 regional, 4 coastal and 1 inland systems. Although the survey covers a limited number of systems, the literature review in previous sections offers additional examples to complete the picture.

Figure 3 provides a graphical summary of the 6 river forcing methods and 4 data sources used in the OOFS listed in Appendix A. In terms of river forcing methods, most systems specify vertical or lateral freshwater fluxes to account for riverine inputs. Only a few of them rely on more sophisticated approaches that use channel extensions within the ocean model or routing schemes from hydrological models to transport the water from the watershed to the coast. Furthermore, none of the global systems surveyed use lateral boundary conditions, likely due to insufficient spatial resolution near river mouths.

In terms of the data sources used in OOFS, what stands out from the survey is the use of in situ data as a primary source in most systems, and climatology either as a primary or fallback source of freshwater discharge. Global systems tend to opt for climatologies in comparison with regional or coastal systems that favour observed data when available, which allows to capture both seasonal and non-seasonal events and their potential local or regional impacts. Only a few regional and inland systems use hydrological models or reanalyses as primary data sources.

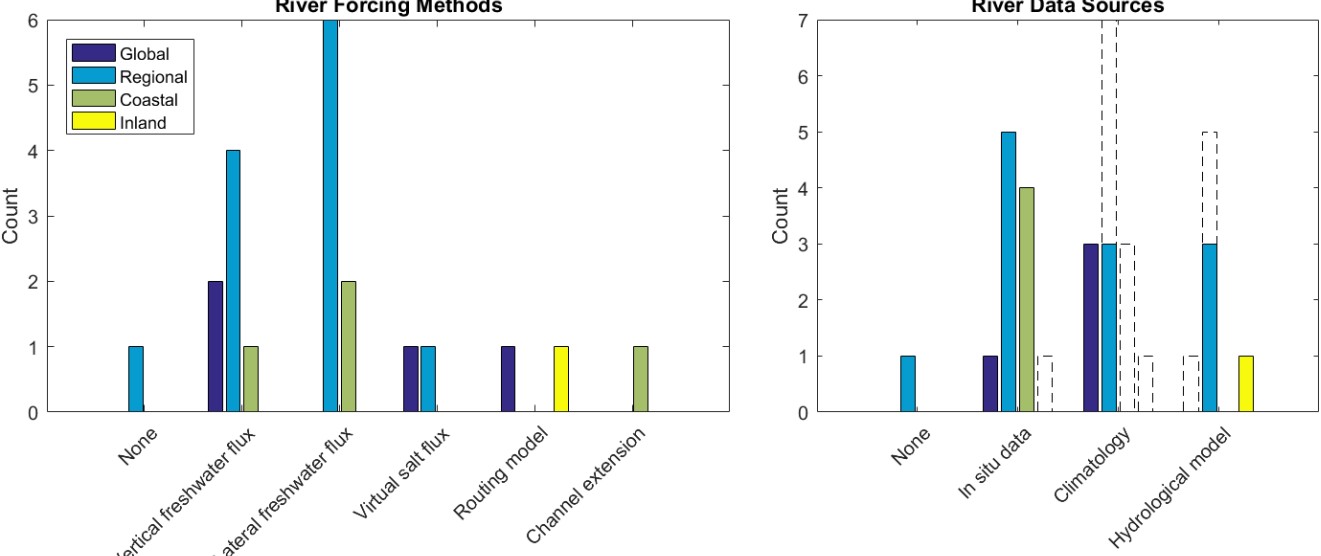

**Figure 3: Graphical summary from a survey on river forcing methods (left panel) and data sources (right panel) used in global, regional, coastal and inland OOFS listed in Appendix A. Coloured bars indicate the primary data sources or methods, whereas dashed bars represent secondary data sources used as a fallback when primary sources are unavailable.**

Additional considerations were also highlighted by the respondents, essential for appropriately representing river inflow in ocean models and addressing challenges such as numerical instabilities and data limitations. For example, spatial smoothing around the river source, or equivalently, optimizing the integration distance for equivalent coastal precipitation may be required to prevent numerical instabilities. Similarly, an increased diffusivity within the surface mixing layer can be implemented to simulate the effects of river inflow. Salinity and temperature of the input freshwater can either be set to zero and to the local SST, respectively, or derived from a combination of real-time gauge data and monthly averages when available. For ungauged areas, river gauge data can be scaled, or additional coastal runoff can be incorporated. In contrast, some systems directly convert precipitation data into river discharges, disregarding hydrological processes and assuming an instantaneous response. In sum, the representation of rivers in OOFS requires careful consideration of various numerical methods, data sources, and modelling approaches. However, some simplifications may limit accuracy in applications requiring high regional precision.

## 5 Summary and recommendations

The assessment of river forcing implementation in OOFS highlights the complexity and challenges of accurately integrating riverine freshwater discharges into ocean models. Despite the growing demand for operational oceanographic products, especially in coastal areas (Ciliberti et al., 2023), OOFS river forcing still faces shortcomings related to the representation of physical processes, data availability, and data quality. The parameterization of river inputs and the interaction between model components, often nonlinear, remain unresolved issues, underscoring the absence of standardized practices for river forcing.

Addressing these gaps requires advancements in model physics, improved spatial and temporal resolution, and enhanced coupling between land, ocean and atmosphere. Furthermore, the incorporation of river flow varies regionally, largely due to differences in the availability and quality of river discharge, salinity and bathymetric datasets, and is further influenced by model scale and resolution. As the demand for reliable coastal forecasts grows, real-time, high-quality river discharge data becomes increasingly pressing. Standardized methodologies and improved integration of riverine parameters—including salinity, temperature, and biogeochemical components—will facilitate seamless watershed-ocean coupling and improve predictions of coastal dynamics, particularly under extreme conditions.

Service evolution roadmaps, such as those outlined by CMEMS, emphasize the need for a better characterization of coastal freshwater exchanges to improve forecasts, especially during severe weather events (Sotillo et al., 2021b). A key step forward involves the progressive replacement of static climatologies with real-time, updated time series (past, present, and forecasts) of river inputs, covering both major and minor or ephemeral streams. Recommendations have been made towards standardized freshwater inputs (and associated river inputs of nutrients and sediment loading), harmonized river forcing approaches, and a more integrated watershed-ocean strategy (Campuzano et al., 2016; Capet et al., 2020; Sobrinho et al., 2021). Additionally, ensuring validated observational error estimates for estuary-mouth forcing, including river discharge and auxiliary variables such as coastal salinity, is crucial for model accuracy (De Mey-Frémaux et al., 2019; Polton et al., 2023). Improved interfaces between coastal monitoring and modelling systems are therefore essential. The FOCCUS project exemplifies progress in addressing these challenges through advancements in hydrological and estuarine modelling, dynamic freshwater inputs, and the integration of AI-driven tools to refine river discharge estimations and coastal system forecasts.

Future efforts must focus on refining model physics, resolution, and coupling strategies to better integrate the land-ocean continuum. Standardized methodologies and integrated high-quality data sources, together with continued interdisciplinary collaboration and technological advancements, will be key to overcoming existing limitations and ensuring more accurate and reliable ocean predictions. Such efforts are critical for improving predictions of coastal dynamics and for fostering a deeper understanding of their implications on global climate and ecosystem functioning.

**Appendix A: Survey on river forcing methods and data sources in current OOFS**

This Appendix presents results of a survey conducted among members of the OceanPredict community in May 2023. The responses are reported in the following tables as given by the participants; nearly no changes were made to each contributed entry, except for a few added references and acronym definitions.

## A.1 Global systems

**Table A.1: Examples of river forcing methods and data sources in global OOFS.**

| System | Institution | Domain(s) | Resolution | Circulation Model | Method for river forcing | Data sources |
|---|---|---|---|---|---|---|
| MOVE/ MRI.COM-G3 [8] (Multivariate Ocean Variational Estimation/ Meteorological Research Institute Community Ocean Model - Global version 3) | Japan Meteorological Agency (JMA)'s Meteorological Research Institute | Global | 1/4° | MRI.COM Ver. 4 | River discharge is expressed as a part of the surface freshwater | Climatology of JRA-55do river runoff data |
| GEOS [9] (NASA Goddard Earth Observing System) | NASA's Global Modeling and Assimilation Office | Global | 25 km – 4 km | MOM6 | GEOS-land component run off, routed to catchments | In situ data, land/catchment model |

---

[8] https://ds.data.jma.go.jp/tcc/tcc/products/elnino/move_mricom-g3_doc.html
[9] https://gmao.gsfc.nasa.gov/GEOS_systems/

| RTOFSv2 [10] (Real-Time Ocean Forecast System) | NOAA's National Centers for Environmental Prediction | Global | 0.08º | HYCOMv2.2 | Rivers are implemented as virtual salt flux at the ocean surface. River runoff is distributed over several ocean grid points around the river source by applying spatial smoothing to spread out the effect of the river and prevent negative salinities due to numerical overshooting. To mimic the river inflow, river freshwater is mixed from the surface down to a depth specified by the user (set to 6 meters in RTOFS). In the grid cells with not-zero river runoff and in the upper layers, river freshwater is mixed within increased vertical diffusivity. Alternatively, rivers can be added directly to the input precipitation fields, which is a better option for a higher (than monthly) frequency river flow data. It is possible to treat rivers (as well as evaporation minus precipitation, E-P) as a mass exchange (not activated in RTOFS). | RTOFS uses global climatology of monthly mean river discharge created at the Naval Research Laboratory (NRL) (Barron and Smedstad, 2002). It provides monthly runoff for 986 rivers. The dataset is based on the Perry (1996) data with corrections and additions derived from: (1) monthly mean streamflow over all years, accessible from the USGS (Wahl et al., 1995); (2) the Global River Discharge (RivDIS) database (Vörösmarty et al., 1998); (3) the Regional, Hydrometeorological Data Network (R-Arcticnet [11]) database provides most of the information ultimately used on rivers flowing into the Arctic, primarily rivers in Russia and Canada. |
|---|---|---|---|---|---|---|

---

[10] https://polar.ncep.noaa.gov/global/about/
[11] http://www.r-arcticnet.sr.unh.edu/

| FOAM-CPL-NWP [12] (Forecast Ocean Assimilation Model, Coupled Numerical Weather Prediction | UK Met Office | Global | 1/4° | NEMO v3.6 | Fresh water runoff from land is input in the surface layer of the ocean with the assumption that the runoff is fresh and at the same temperature as the local sea surface temperature. An enhanced vertical mixing of $2 \times 10^{-3}$ m$^2$s$^{-1}$ is added over the top 10 m of the water column at runoff points to mix the runoff vertically and avoid instabilities associated with very shallow fresh layers at the surface (Storkey et al., 2018). | Climatological river runoff fields were derived by Bourdalle-Badie and Treguier (2006) based on estimates given in Dai and Trenbert (2002) (Blockley et al., 2014) |

405

## A.2 Regional systems

**Table A.2: Examples of river forcing methods and data sources in regional OOFS.**

| System | Institution | Domain(s) | Resolution | Circulation Model | Method for river forcing | Data sources |
|---|---|---|---|---|---|---|
| MOVE/MRI.COM-NP/JPN [13] (Multivariate Ocean Variational Estimation/ | Japan Meteorological Agency (JMA)'s Meteorological | North Pacific | 2 km - 10 km | MRI.COM Ver. 5 | River discharge is expressed as a part of the surface freshwater | Climatology of JRA-55do river runoff data |

---

[12] https://www.metoffice.gov.uk/services/data/met-office-data-for-reuse/model
[13] https://www.data.jma.go.jp/kaiyou/data/db/kaikyo/knowledge/move_jpn/system.html

| | | | | | | |
|---|---|---|---|---|---|---|
| Meteorological Research Institute Community Ocean Model – North Pacific/ Japan) | Research Institute | | | | | |
| TOPAZ[14] | Norwegian's Nansen Environmental and Remote Sensing Center (NERSC) | Arctic and Nordic Seas | 12 km | HYCOM | Removal of salt from the surface (an ellipse around the river mouth) and barotropic water flux. We use nutrients (N, P and Si) from the globalNEWS model and scale them by river discharge. | Swedish Meteorological and Hydrological Institute (SMHI) (Arctic-HYPE and E-HYPE), GRACE satellite for Greenland mass loss and a home-made climatology for Greenland surface mass balance. |
| eSA-Marine[15] | South Australian Research and Development Institute | South Australian Gulfs and Shelf | 2.5 km and 0.5 km | ROMS | None, intermittent river input is usually weak to non-existent. | None |
| DMI HYCOM-CICE[16] | Danish Meteorological Institute (DMI) | Arctic and Atlantic Oceans | 4-10 km: ~5 km throughout Arctic and northern Atlantic | HYCOM + CICE fully coupled using Earth System Modeling Framework | River forcing is converted to monthly means precipitation equivalents [m/s] for ~50,000 river-runoff outlets and distributed to the nearest coastal | River forcing is taken from various sources using a dataset from the Geological Survey of Denmark and Greenland (Mankoff et al., 2020), |

---

[14] https://nersc.no/en/products-and-services/analysis-tools-and-models/ocean-models/
[15] https://pir.sa.gov.au/research/services/esa_marine/about_esa-marine
[16] https://ocean.dmi.dk/models/hycom.uk.php

| | | | | | | |
|---|---|---|---|---|---|---|
| | | | | (ESMF) coupler. CICE runs on a subset of the full HYCOM domain | model grid point(s) (Ponsoni et al., 2023). | converted to monthly means precipitation equivalents [m/s] |
| DKSS[17] (Danish Storm Surge System) | Danish Meteorological Institute | North Sea - Baltic Sea, with multiple nested subdomains | 3 nautical miles (coarsest) to 0.1 nautical mile (finest) | HBM (Hiromb-Baltic Model) | River forcing is treated as a freshwater flux into coastal grid cells. Water temperature equal to receiving cell (river temperature data not used) with 0°C as lower limit to avoid instantaneous freezing. | European hydrological model E-HYPE3, from which an annual plus a calendar day ~30y climatology has been derived and used as a back-up for a daily forecast. The forecast model is run by the Swedish Hydrological and Meteorological Institute, and the day-to-day service comes with an annual fee. |
| IBI Near-Real-Time[18] | Iberia Biscay Irish (IBI) Sea – Monitoring Forecasting Center | European Atlantic façade (the Iberia-Biscay-Ireland zone): Lat: from 26N to 56N, Lon: | 1/36°, Surface and 3D fields (50 vertical levels) | NEMO v3.6 | Freshwater river discharge inputs are implemented as lateral open boundary conditions for the main 33 rivers of the IBI area. The system also incorporates an extra coastal runoff rate (derived from the Dai and Trenberth (2002) climatology, on a monthly | Data come from different sources, depending on their availability, in the following order: (1) Model data: SMHI hydrologic model; (2) Monthly climatological data taken from GRDC, French "Banque |

---

[17] https://opendatadocs.dmi.govcloud.dk/Data/Forecast_Data_Storm_Surge_Model_DKSS#:~:text=DKSS%20is%20DMI%27s%20forecast%20model,ice%20thickness%20and%20ice%20concentration

[18] https://marine.copernicus.eu/about/producers/ibi-mfc

| | | | | | | |
|---|---|---|---|---|---|---|
| | | from 19W to 5E | | | basis), which makes the IBI forcing consistent with the ones imposed in the parent Copernicus Marine GLOBAL system. | Hydro" [19] dataset, Copernicus Marine Service and EMODnet. |
| IBI Multi-Year[20] | Iberia Biscay Irish (IBI) Sea – Monitoring Forecasting Center | European Atlantic facade (the Iberia-Biscay-Ireland zone): Lat: from 26N to 56N, Lon: from 19W to 5E | 1/12°, Surface and 3D fields (50 vertical levels) | NEMO v3.6 | Same as IBI-NRT, but with an additional river (LAGAN) | Data come from different sources, depending on their availability, in the following order: (1) In-situ data: daily measurements from Copernicus Marine Service, EMODnet or national web sites; (2) Model data: SMHI hydrologic model. |
| CBEFS [21] (Chesapeake Bay Environmental Forecast System) | Virginia Institute of Marine Science | Chesapeake Bay | 600 m x 600 m | ROMS | Freshwater - Real time USGS river gauge data is scaled to better represent total freshwater inflows over a larger area based on a watershed model. The scaled discharge is then disaggregated into the main river inflow and smaller streams based on proportions developed from the watershed model. The | In situ gauge data. Hindcast watershed model information. Artificial Neural Networks. |

---

[19] http://www.hydro.eaufrance.fr/
[20] https://marine.copernicus.eu/about/producers/ibi-mfc
[21] https://www.vims.edu/research/products/cbefs/

| | | | | | | |
|---|---|---|---|---|---|---|
| | | | | | forecast is a simple autoregressive model based on the past few days. Riverine Biogeochemistry - Inputs are specified using Artificial Neural Network AI models based on the discharge and date, which recreate what the watershed model would have predicted had the current and forecast conditions been simulated by the watershed model. Temperature - Water temperature is specified using a combination of real time gauge data and monthly averages depending on what is available. | |
| DREAMS [22] (RIAM Real-Time Ocean Forecasting) | Kyushu University's Research Institute for Applied Mechanics (RIAM) | East Asian marginal seas | 0.3 – 22 km | RIAM Ocean Model | Coastal precipitation is directly converted into the amount of river discharges. The integration distance was optimized by using model Green's functions (Hirose, 2011). | Grid point value (GPV) precipitation data of Japan Meteorological Agency (JMA) |

---

[22] https://dreams-c1.riam.kyushu-u.ac.jp/vwp/

| FOAM-AMM15 [23] (Forecast Ocean Assimilation Model– Atlantic Margin model 1.5km) | UK Met Office | Northwest European Shelf Seas | 1.5 km | NEMO v3.6 | For each river input location, a daily freshwater flux is assigned, with depth determined by the average ratio of runoff to tidal range (as per the estuary classifications of Cameron and Pritchard, 1963). The runoff temperature is assumed to align with the local sea surface temperature (SST), as the climatology does not include temperature data (Graham et al., 2018). | River runoff is primarily derived from a daily climatology of gauge measurements averaged for 1980–2014. UK data were processed from raw data provided by the Environment Agency, the Scottish Environment Protection Agency, the Rivers Agency (Northern Ireland), and the National River Flow Archive (gauge data were provided by Sonja M. van Leeuwen, CEFAS, Lowestoft, UK, personal communication, 2016). For major rivers that were missing from this data set (e.g. along the French and Norwegian coasts), data have been provided from an earlier climatology (Vörösmarty et al., 2020; Young and Holt, 2007), based on a daily climatology of gauge data averaged for the period 1950–2005 (Tonani et al., 2019). |

---

[23] https://www.metoffice.gov.uk/services/data/met-office-data-for-reuse/model

| | | | | | | |
|---|---|---|---|---|---|---|
| FOAM-AMM7 [24] (Forecast Ocean Assimilation Model– Atlantic Margin model 7km) | UK Met Office | Northwest European Shelf Seas | 7 km | NEMO v3.6 (coupled to ERSEM 20.10 for biogeochemistry) | For each river input location, a daily freshwater flux is assigned, with depth determined by the average ratio of runoff to tidal range (as per the estuary classifications of Cameron and Pritchard, 1963). The runoff temperature is assumed to align with the local sea surface temperature (SST), as the climatology does not include temperature data (Graham et al., 2018). | Daily timeseries of river discharge, nutrient loads (nitrate, phosphate, silicate, ammonia), alkalinity (bioalkalinity, dissolved organic carbon) and oxygen were produced from an updated version of the river dataset used in Lenhart et al. (2010), combined with climatology of daily discharge data from the Global River Discharge Database (Vörösmarty et al., 2020) and from data prepared by the Centre for Ecology and Hydrology as used by Young and Holt, 2007. The climatology has an annually-varying component until 2018 to account for historic changes in nutrient loads, values for 2018 are used as a climatology in the operational system (Kay et al., 2020). |
| DOPPIO[25] and MARACOOS[26] (Mid-Atlantic | Rutgers University | Northeast USA and | 7 km | ROMS | Discharge is introduced as volume flux divergence (method LwSrc in ROMS) at | Daily USGS discharge data are scaled for ungauged portions of the watershed based on the |

---

[24] https://www.metoffice.gov.uk/services/data/met-office-data-for-reuse/model
[25] https://gmd.copernicus.org/articles/13/3709/2020/
[26] https://maracoos.org/

| | | Nova Scotia, Canada | | | 27 point sources in model cells adjacent to the coast. | statistics of a 10-year hydrological model analysis. |
|---|---|---|---|---|---|---|
| Regional Association Coastal Ocean Observing System) | | | | | | |

## A.3 Coastal systems

**Table A.3: Examples of river forcing methods and data sources in coastal OOFS.**

| System | Institution | Domain(s) | Resolution | Circulation Model | Method for river forcing | Data sources |
|---|---|---|---|---|---|---|
| DFO's Port Ocean Prediction Systems[27] | Government of Canada's Department of Fisheries and Oceans (DFO) | Kitimat Fjord, Vancouver Harbour, Lower Fraser River, St Lawrence Estuary, Port of Canso, Saint John harbour | 20 – 200 m | NEMO 3.6 | NEMO's runoff feature for some rivers, and a SSH open boundary condition for others | Gauge data (from Environment and Climate Change Canada, ECCC) where available, climatology elsewhere |
| CIOPS [28] (Coastal Ice-Ocean Prediction System) | Environment and Climate Change Canada (ECCC) | East/West + SalishSea500 | 1/36° + 500m for SS500 | NEMO 3.6 | Same as DFO port models | Gauge data for Fraser River, climatology elsewhere |

[27] https://publications.gc.ca/site/eng/9.905464/publication.html

[28] https://eccc-msc.github.io/open-data/msc-data/nwp_ciops/readme_ciops_en/

| | | | | | | |
|---|---|---|---|---|---|---|
| FANGAR BAY[29] | Universitat Politècnica de Catalunya | Ebro Delta | 350m / 70m | COAWST (ROMS/ SWAN) | Climatological freshwater from Ebro River | In situ data |
| NARF [30] (Northern Adriatic Reanalysis and Forecasting system) | Istituto Nazionale di Oceanografia e di Geofisica Sperimentale | Northern Adriatic Sea (Mediterranean Sea) | 1/128° (~750 m) | MITgcm-BFM (coupled hydrodynamic-biogeochemical) | The downstream end of the rivers flowing into the basin is simulated as a narrow channel: one or two cells in the horizontal direction and a few vertical levels. Freshwater discharge rates from NRT data or climatologies are converted into horizontal velocities (the section of the riverbed is known) and applied as lateral open boundary conditions. Salinity is constant (5 PSU), temperature has a yearly sinusoidal cycle (maxima and minima in summer and winter, respectively) and biogeochemical concentrations are derived from literature/climatologies. | In-situ NRT discharge data for the Po River (main contributor), climatologies for the others (with sinusoidal modulation: maxima in spring/fall, minima in summer/winter). Daily frequency. |

---

[29] https://doi.org/10.5194/egusphere-egu24-11220
[30] https://medeaf.ogs.it/got

## A.4 Inland systems

410

**Table A.4: Example of river forcing methods and data sources in inland OOFS.**

| System | Institution | Domain(s) | Resolution | Circulation Model | Method for river forcing | Data sources |
|---|---|---|---|---|---|---|
| WCPS [31] (Water Cycle Prediction System) | Environment and Climate Change Canada (ECCC) | Great-Lakes+ Northwest Atlantic (NWA) | 1/36° + 1km | NEMO 3.6 | Fully coupled hydrologic model for GL, climatology for NWA | Hydrological model uses gauge data |

---

[31] https://eccc-msc.github.io/open-data/msc-data/nwp_wcps/readme_wcps_en/

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

**Competing interests**

The contact author has declared that none of the authors has any competing interests.

**Data and/or code availability**

Data/code availability is not applicable to this article as no new data/code were created or analysed in this study.

775 **Authors contribution**

Pascal Matte: Conceptualization, Investigation, Writing – original draft preparation, Writing – review and editing. John Wilkin: Writing – review and editing. Joanna Staneva: Writing – review and editing.

**Acknowledgements**

The authors wish to thank Kristen Wilmer-Becker and members of the OceanPredict community who participated in the survey
780 conducted in May 2023 on the status of implementation of river forcing in current OOFS, as well as three anonymous reviewers for their constructive comments on this manuscript.