# Peer review of "The Representation of Rivers in Operational Ocean Forecasting Systems: A Review"

_State of the Planet, 2024_

## Referee Comment (RC2)

**TITLE** - The Role of Rivers in Ocean Forecasting

**RECOMMENDATION**: Major revisions

This paper aims at providing a review of the river release representation in ocean modeling, spannnig from global to coastal scales.
It offers a valuable contribution to ocean modeling developers and practitioners, as it assesses current advancements and offers recommendations for the next generation modeling of the global coastal ocean to more accurately account for riverine inputs.

I encourage the authors to make additional efforts to deliver a comprehensive and detailed overview of the current state of the field, ensuring it serves as a valuable reference for the community.

As it stands, the manuscript is lacking in several key topics. It would greatly benefit from incorporating a broader range of relevant studies and addressing open issues that deserve reporting and discussion.

*Major comments:*

Title: I suggest to modify the title to clarify the the paper aims at providing a review of the state of art of river release representation within OOFS

Line 52-53: The "point source input" here mentioned is not a rigorous definition. More precise, the river release entering as surface point sources affects the vertical velocity surface boundary condition of the free surface equation (i.e. the vertically integrated continuity equation, not the continuity equation itself) and the surface boudary conditions for the diffusive heat and salt fluxes (Beron Vera 1999)

Line 70: "to account for baroclinic flow", I'd suggest to replace with "to account for baroclinic and barotropic flows"

Line 75: Figure 1 caption. Which EBM does the caption refer to? Please include a reference to the model

Line 97-98: please detail more the relevant result found out by Bao et al 2022 by comparing 2way versus linked approach.

Line 98-99: this sentence is too concise. Moreover the seamless river-sea continuum modeling deserves a specific additional section, e.g. Section 2.4

Add discussion and references on the topics below:
> **-The Influence of Seasonal and Non-Seasonal River Release on Stratification and Sea Level Variability:** It is important to discuss how variations in river release can impact both stratification and sea level changes. Relevant studies include in addition to the already mentioned Chandanpurker et al 2022:
> > o Zhang, Y. J., Ye, F., Stanev, E. V., & Grashorn, S. (2016). Seamless cross-scale modeling with SCHISM. *Ocean Modelling*, *102*, 64-81.

- o Giffard, P., Llovel, W., Jouanno, J., Morvan, G., & Decharme, B. (2019). Contribution of the Amazon River discharge to regional sea level in the tropical Atlantic ocean. *Water*, *11*, 2348. https://doi.org/10.3390/w11112348
  - o Piecuch, C. G., Bittermann, K., Kemp, A. C., Ponte, R. M., Little, C. M., Engelhart, S. E., & Lentz, S. J. (2018). River-discharge effects on United States Atlantic and Gulf coast sea-level changes. *Proceedings of the National Academy of Sciences*, *115*(30), 7729-7734.
  - o Piecuch, C. G., & Wadehra, R. (2020). Dynamic sea level variability due to seasonal river discharge: A preliminary global ocean model study. *Geophysical Research Letters*, *47*(4), e2020GL086984.
  - o Verri, G., Pinardi, N., Oddo, P., Ciliberti, S. A., & Coppini, G. (2018). River runoff influences on the Central Mediterranean overturning circulation. Climate dynamics, 50(5-6), 1675-1703

- **Unstructured Modeling of the River-Sea Continuum:** This approach offers various advantages, including alleviating the challenges associated with prescribing river salinity because it can be set equal to zero at the head of an estuary solved by an unstructured grid. A dedicated section discussing this topic is warranted. Key references for this discussion include, in addition to the already mentioned Zhang et al 2016: Le Bars et al 2016, Maicu et al (2021), Bellafiore et al (2021), Vallaeys et al. (2018), Vallaeys et al. (2021), and Verri et al. (2023), Bonamano et al (2024), among many others.

- **Machine Learning Approaches to Estimate Riverine Release.** Key references to consider include studies that highlight successful machine learning applications in hydrology and oceanography. Regarding the salinity at river mouths some references are provided below:
  - o Fang, Y., wei Chen, X., Cheng, N.-S., 2017. Estuary salinity prediction using a coupled GA-SVM model: A case study of the Min River Estuary, China. Water Sci. Technol.: Water Supply 17, 52–60.
  - o Qiu, C., Wan, Y., 2013. Time series modeling and prediction of salinity in the Caloosahatchee River Estuary. Water Resour. Res. 49 (9), 5804–5816. http: //dx.doi.org/10.1002/wrcr.20415, arXiv:https://agupubs.onlinelibrary.wiley.com/ doi/pdf/10.1002/wrcr.20415, URL: https://agupubs.onlinelibrary.wiley.com/doi/ abs/10.1002/wrcr.20415.
  - o Guillou, N., Chapalain, G., Petton, S., 2023. Predicting sea surface salinity in a tidal estuary with machine learning. Oceanologia 65 (2), 318–332. http://dx.doi.org/10. 1016/j.oceano.2022.07.007, URL: https://www.sciencedirect.com/science/article/ pii/S0078323422000835.
  - o Qi, S., He, M., Bai, Z., Ding, Z., Sandhu, P., Zhou, Y., Namadi, P., Tom, B., Hoang, R., Anderson, J., 2022b. Multi-location emulation of a process-based salinity model using machine learning. Water 14 (13), http://dx.doi.org/10.3390/w14132030, URL: https://www.mdpi.com/2073-4441/14/13/2030
  - o Saccotelli, L., Verri, G., De Lorenzis, A., Cherubini, C., Caccioppoli, R., Coppini, G., Maglietta, R., 2024. Enhancing estuary salinity prediction: a

Machine Learning and Deep Learning based approach. *Applied Computing and Geosciences*

- o Maglietta, R., Verri, G., Saccotelli, L., De Lorenzis, A., Cherubini, C. Caccioppoli, R., Dimauro, G., Pinardi, N., Coppini, G. (2024) Advancing Estuarine Box Modeling: a Novel Hybrid Machine Learning and Physics-Based Approach. Environmental Modelling and Software

Regarding the water level along estuaries:

- o Sampurno, J., Vallaeys, V., Ardianto, R., and Hanert, E.: Integrated hydrodynamic and machine learning models for compound flooding prediction in a data-scarce estuarine delta, Nonlin. Processes Geophys., 29, 301–315, https://doi.org/10.5194/npg-29-301-2022, 2022.

- **Estimation of River Temperature and Its Minor Role:** The estimation of river temperature should be acknowledged in the context of riverine release even if it is not the primary factor influencing oceanographic processes

*Minor comments:*
Line 10: I'd mention also the subsurface water discharge

Line 40: The first time the acronym OOFS is mentioned, it should be spelled out in full.

Line 48-49: the prescribed salinity values at river mouths are tipically based on costant annual/monthy values which are the result of sensitivity tests and/or in situ campaigns *(Verri, G., Pinardi, N., Oddo, P., Ciliberti, S. A., & Coppini, G., 2018. River runoff influences on the Central Mediterranean overturning circulation. Climate dynamics, 50(5-6), 1675-1703)*

Line 75: Figure 1 caption. Which EBM does the caption refer to? Please include a reference to the model

Line 111-112: I believe here you should refer to Verri et al 2018 rather than Verri et al 2021. Sensitivity tests by Verri et al 2018 demonstrate that a more realistic estimates of riverine inputs would produce a more accurate representation of coastal (plume) to basin wide circulation and dynamics (dense water formation, overturing circulation cells, water exchange at the straits …)

Line 192: "However, strong land-sea differences in microwave emissivity make satellite observations unreliable within some 70 km of the coast" I belive the water turbidity is the main limit to be mentioned here

Line 194-195: The satellite retrieved salinity close to the river mouth is a crucial chalenge and more recent studies should be mentioned and briefly discussed
*e.g. Medina et al 2020; Sakai et al 2021, Chen et al 2017*

Section 3.1.2: a missing reference in the list is the recent database of climatological runoff for the Adriatic rivers provided by *Aragão, L., Mentaschi, L., Pinardi, N., Verri, G., Senatore, A., & Di Sabatino, S. (2024). The freshwater discharge into the Adriatic Sea revisited. Frontiers in Climate*

Section 3.3 all theOOFS should be references through links to their web pages /pubblications

---

## Author Comment (AC1)

**RESPONSE TO REVIEWER #1'S COMMENTS**

**General comments**

The authors provide an overview of the role of river forcing in ocean models, with an emphasis on current status of Operational Ocean Forecasting Systems (OOFS). This offers an accessible introduction and status update. providing a valuable reference to those working on OOFS and to a wider community for whom the inner-workings of OOFS will be less familiar. The review paper provides a fair reflection of the current state.

*Response: We thank the Reviewer for this positive and constructive feedback on our manuscript. A detailed response to specific comments is provided below, using italic text in blue.*

**Specific comments**

1. Paper title: Given the emphasis on OOFS, consider a more specific title e.g. "The Representation of Rivers in Operational Ocean Forecasting Systems". This better represents the paper content in my view. Consider also updating the Abstract to be clear on scope. or example in Line 13, the authors suggest that *"This paper provides an overview of recent advances in river modelling"* which might suggest a detailed review of hydrological process representation, whereas I think the paper rather more provides "an overview in recent approaches to representing coastal river discharges and processes in ocean models".

*Response: We agree with the Reviewer. The title was modified to better reflect the paper content as suggested: "The Representation of Rivers in Operational Ocean Forecasting Systems: A Review". The abstract was also modified as suggested. The sentence now reads: "This paper provides an overview of recent approaches to representing coastal river discharges and processes in ocean models, with a particular focus on estuaries."*

2. In general review of freshwater impacts on ocean prediction in opening paragraph, it would be worth including example references from Bay of Bengal, as an area with significant sensitivity to freshwater influence (e.g. Jana et al 2015 - https://doi.org/10.1016/j.csr.2015.05.001), and a body of literature on resulting influence of BoB freshwater and barrier layers effects on TC and broader monsoon development.

*Response: We thank the reviewer for this suggestion. We added the following text in a new Section 2.1 (Capturing seasonal and non-seasonal river variability) rather than in the Introduction: "The Bay of Bengal is one example where the inclusion of seasonal river discharges and salinity in regional model simulations significantly improved the representation of sea surface temperatures, near-surface salinity, stratification, mixed-layer depth and barrier-layer thickness, leading to a better simulation of the formation, progression and dispersion of the freshwater plume (Jana et al., 2015)."*

*In the introduction, only minor additions were made, for example (underlined):*
*"At coarse scales that cannot resolve the estuarine dynamics, but even at finer scales in some cases, river outlets are often represented in a simplistic way, with climatological runoff and zero or constant salinity values, implicitly neglecting estuarine mixing and exchange as well as seasonal and non-seasonal variability (Sun et al., 2017; Verri et al., 2020; Verri et al., 2021)."*

3. Paper structure:

- In describing paper contents on line 40, please sign-post later sections more precisely - i.e. "Section 2 reviews...." etc.

  *Response: We agree. The structure of the paper is now described precisely in the last paragraph of the Introduction.*

- I would recommend addition of a brief Section 4: Discussion & Conclusions section that highlights some recommendations for the community, and highlights gaps in capabilities and knowledge to address (see also comment below).

  *Response: A new Section 5 (Summary and recommendations) was added, discussing current limitations in OOFS and providing recommendations on ways forward, as follows:*

  *"The description of the status of implementation of river forcing in OOFS revealed the complexity and limitations associated with appropriately treating riverine freshwater discharges into ocean models. Despite a growing demand for operational oceanographic products and services, especially in coastal areas (Ciliberti et al., 2023), OOFS still suffer from limitations with respect to (1) the representation of the physical processes and (2) data availability and quality. How river forcing inputs are defined or parameterized and how model components interact, often non-linearly, remain challenging questions, as they point to a lack in the definition of standard practices regarding river forcing. An enhanced representation of rivers in OOFS must go through improvements in model physics, appropriate spatial and temporal resolutions, and coupling between land, ocean and atmosphere. The difficulty in incorporating river flow also varies by geographic region, especially with respect to the availability and quality of river discharge, salinity and bathymetric datasets. It is further modulated as a function of model scale and resolution (whether one deals with a global, regional, or coastal model configuration).*

  *Sotillo et al. (2021b) highlighted that service evolution roadmaps, such as the CMEMS guidelines, need to include a better characterization of the land boundary, especially of the coastal freshwater exchanges, to improve forecasts particularly under severe weather conditions. This includes real-time updated time series (past, present, forecasts) of river inputs for both major and minor or ephemeral streams as a progressive replacement of static climatologies. Recommendations towards standardized inputs of freshwater (and associated river inputs of nutrients and sediment loading), homogenized river forcing approaches, and a more integrated watershed-ocean strategy are being made (Campuzano et al., 2016; Capet et al., 2020; Sobrinho et al., 2021). Validated observational error estimates must also be a priority to ensure accurate estuary-mouth forcing (De Mey-Frémaux et al., 2019; Polton et al., 2023), for river discharge as well as auxiliary variables, such as coastal salinity. As such, improved interfaces between coastal monitoring and modelling systems are required. The FOCCUS project exemplifies efforts to address these challenges, particularly through its advancements in hydrological and estuarine modelling, dynamic freshwater inputs, and integration of AI-driven tools to enhance river discharge estimations and coastal system forecasts.*

  *This study underscores the complexity and importance of accurately representing riverine freshwater discharges in OOFS. The challenges lie not only in the variability of river forcing*

*methods but also in the diversity and quality of data sources available across different geographical and temporal scales. Advancing the representation of riverine processes requires improvements in model physics, resolution, and coupling strategies to better integrate the land-ocean continuum. As the demand for reliable coastal forecasts grows, the need for real-time, high-quality river discharge data becomes increasingly pressing. Standardized methodologies and enhanced integration of riverine parameters, including salinity, temperature, and other biogeochemical components, will support seamless coupling between watershed and ocean models. Such efforts are critical for improving predictions of coastal dynamics, particularly under extreme weather conditions, and for fostering a deeper understanding of their implications on global climate and ecosystem functioning."*

- Recommend to move the detailed Tables 1-4 into Appendix materials, but draw out any key themes/similarities/differences in the main manuscript as shorter (more digestible) Section 3.3.

    *Response: We agree. Tables 1-4 were moved to a new Appendix A and section 3.3 was moved to a new Section 4 and expanded as follows:*

    *"Figure 3 provides a graphical summary of the 6 river forcing methods and 4 data sources used in the OOFS listed in Appendix A. In terms of river forcing methods, most systems specify vertical or lateral freshwater fluxes to account for riverine inputs. Only a few of them rely on more sophisticated approaches that use channel extensions within the ocean model or routing schemes from hydrological models to transport the water from the watershed to the coast. Furthermore, global systems inventoried in the survey do not use lateral boundary conditions, possibly due to a lack of spatial resolution near river mouths.*

    *In terms of the data sources used in OOFS, what stands out from the survey is the use of in situ data as a primary source in most systems, and climatology either as a primary or fallback source of freshwater discharge. Global systems tend to opt for climatologies in comparison with regional or coastal systems that favour observed data when available, which allows to capture the non-seasonal events and their potential local or regional impacts. Data from hydrological models or reanalyses were only given as primary data sources in a few regional and inland systems.*

[Figure]

*Figure 3: Graphical summary from a survey on river forcing methods (left panel) and data sources (right panel) used in global, regional, coastal and inland OOFS listed in Appendix A. Coloured bars indicate the primary data sources or methods, whereas dashed bars represent secondary data sources used as a fallback when primary sources are unavailable.*

*Additional considerations were also highlighted by the respondents, essential for appropriately representing river inflow in ocean models and addressing challenges such as numerical instabilities and data limitations. For example, spatial smoothing around the river source, or equivalently, optimizing the integration distance for equivalent coastal precipitation may be required to prevent numerical instabilities. Similarly, an increased diffusivity within the surface mixing layer can be implemented to simulate the effects of river inflow. Salinity and temperature of the input freshwater can either be set to zero and to the local SST, respectively, or derived from a combination of real-time gauge data and monthly averages when available. To account for ungauged areas, river gauge data can be scaled, or additional coastal runoff can be included. In contrast, some systems directly convert precipitation data into river discharges, disregarding hydrological processes and assuming an instantaneous response.*

*In sum, the representation of rivers in OOFS requires careful consideration of various numerical methods, data sources, and modelling approaches. However, certain choices or simplifications may pose limitations for applications demanding high accuracy in specific regions."*

4. Recommendations and additional commentary

- While the paper compiles a list of various potential data sources and products, I am missing a particular narrative on relative strengths/weaknesses/limitations of the various approaches.

  *Response: See response above and to next comment.*

- This could partly be mitigated by slightly more expansion where relevant and available on reference discussions of data quality. As a specific, but not limited, example, GloFAS-ERA5 is

highlighted as an operational discharge product in Section 3.1.3, but without discussion of its quality, or recommendation of its value for OOFS.

*Response: Our objective in this paper is not to assess the quality or value of a given product or dataset, but rather to make an inventory of existing current approaches and available datasets. A detailed gap analysis and identification of ways forward will be the topic of a follow-up contribution. However, in this paper we document what are the preferred options adopted in many studies for inputting river discharge in OOFS. We also modified the manuscript as follows:*

*We added in Section 3.1.2 on River Databases: "Furthermore, a detailed comparative assessment of these various data sources is still lacking."*

*We added in Section 3.1.3 on Operational river discharge products: "The FOCCUS project (Forecasting and Observing the Open-to-Coastal Ocean for Copernicus Users, foccus-project.eu) further enhances operational hydrological models by addressing the land-ocean continuum through improved river runoff estimations and the development of advanced coupling between hydrological and coastal ocean models. FOCCUS builds on existing pan-European hydrological frameworks, such as E-HYPE and LISFLOOD, to provide dynamic freshwater inputs, including nutrient and inorganic matter transport. Additionally, the project integrates novel AI techniques to optimize estuarine modeling and freshwater forcing for coastal systems. These innovations directly contribute to refining CMEMS and supporting all European coastal services with more accurate and seamless coastal monitoring and forecasting capabilities.*

*In some instances, the regional products may appear to be the preferred option for some regional or local studies, as they were designed to specifically represent the hydrological characteristics of a given area, sometimes with higher resolution and accuracy. However, a global solution is attractive in data scarce areas and where consistency between discharge products and across all forcing variables is required over large domains (Polton et al., 2023)."*

*Finally, we included a new Section 5 (Summary and recommendations) where a summary of challenges and limitations is provided, followed by a few recommendations on ways forward.*

- While Section 3.3 provides useful reference detail on OceanPredict community approaches, the authors could help synthesise the details from the Table to better reflect in qualitative discussion the various approaches - e.g. numbers using climatologies, river models etc.

  *Response: The Section 4 (formerly Section 3.3) now includes a qualitative discussion and a new Figure 3 synthetizing results from the survey (also copied in a response above).*

4. Authors might also consider referencing Polton et al 2022 (https://doi.org/10.5194/gmd-16-1481-2023) as additional example of practical guide on implementing freshwater OBC inputs to ocean models (see their Section 3.7).

*Response: We thank the reviewer for this suggestion. Reference from Polton et al. (2023) was added. More specific details found in this review paper were also added, as follows:*

- *In Section 2.2, we added: "Additional subtleties arise for large rivers or deltas, where the coastal source points need to be spread laterally to avoid numerical instabilities if inflow values are locally too large (Polton et al. 2023)."*
- *In section 3.1.2, we added a reference to the river database found in the CORE.v2 dataset (Large and Yeager, 2009): "A global freshwater budget is included in the CORE.v2 datasets that have an accompanying database for continental runoff from rivers, groundwater and icebergs, estimated from continental imbalances between precipitation, evaporation and storage, then distributed between bordering ocean basins based on river routing schemes and flow estimates (Large and Yeager, 2009)."*
- *In Section 3.1.3 we added: "In some instances, the regional products may appear to be the preferred option for some regional or local studies, as they were designed to specifically represent the hydrological characteristics of a given region, sometimes with higher resolution and accuracy. However, a global solution is attractive in data scarce areas and where consistency between discharge products and across all forcing variables is required over large domains (Polton et al., 2023)."*
- *In Section 5 we added a citation to Polton et al.: "Validated observational error estimates must also be a priority to ensure accurate estuary-mouth forcing (De Mey-Frémaux et al., 2019; Polton et al., 2023), for river discharge as well as auxiliary variables, such as coastal salinity."*

---

## Author Comment (AC2)

**RESPONSE TO REVIEWER #2'S COMMENTS**

**TITLE** - The Role of Rivers in Ocean Forecasting

**RECOMMENDATION**: Major revisions

This paper aims at providing a review of the river release representation in ocean modeling, spanning from global to coastal scales.

It offers a valuable contribution to ocean modeling developers and practitioners, as it assesses current advancements and offers recommendations for the next generation modeling of the global coastal ocean to more accurately account for riverine inputs.

I encourage the authors to make additional efforts to deliver a comprehensive and detailed overview of the current state of the field, ensuring it serves as a valuable reference for the community.

As it stands, the manuscript is lacking in several key topics. It would greatly benefit from incorporating a broader range of relevant studies and addressing open issues that deserve reporting and discussion.

*Response: We thank the reviewer for these valuable comments. A detailed response to specific comments is provided below, using italic text in blue.*

***Major comments:***

Title: I suggest to modify the title to clarify the the paper aims at providing a review of the state of art of river release representation within OOFS

*Response: We modified the title as follows: "The representation of rivers in operational ocean forecasting systems: A review".*

Line 52-53: The "point source input" here mentioned is not a rigorous definition. More precise, the river release entering as surface point sources affects the vertical velocity surface boundary condition of the free surface equation (i.e. the vertically integrated continuity equation, not the continuity equation itself) and the surface boundary conditions for the diffusive heat and salt fluxes (Beron Vera 1999)

*Response: We made the following changes to the text (underlined): "A first approach, referred to as a point source input, adds a term of freshwater flux, entering as surface point sources into one or more layers of the model, to the divergence of flow in the vertically integrated continuity equation, with no associated velocity profile. It affects the vertical velocity surface boundary condition of the free surface equation, and the surface boundary conditions for the diffusive heat and salt fluxes."*

Line 70: "to account for baroclinic flow", I'd suggest to replace with "to account for baroclinic and barotropic flows"

*Response: Corrected.*

Line 75: Figure 1 caption. Which EBM does the caption refer to? Please include a reference to the model

*Response: The reference was already included as a footnote. We also added this information in the caption: "Schematic diagram of the estuary box model (EBM) implemented in the Community Earth System Model (CESM) (Sun et al., 2017)."*

Line 97-98: please detail more the relevant result found out by Bao et al 2022 by comparing 2way versus linked approach.

*Response: We added the following sentence: "In a case study of Hurricane Florence, Bao et al. (2022) achieved significant improvement in simulated water levels (20%-40% at the head of Cape Fear River Estuary) during the post-hurricane period by using a two-way coupled model, compared to a stand-alone and linked (one-way coupled) approach. This higher performance can be largely explained by the momentum exchanges at the model interfaces, capturing the nonlinear effects between runoff and residual water level from the ocean."*

Line 98-99: this sentence is too concise. Moreover the seamless river-sea continuum modeling deserves a specific additional section, e.g. Section 2.4

*Response: We decided to move this sentence (and additional related content) to Section 2.3 (Freshwater input in high resolution models) instead, as it relates more closely to the development of high resolution models than to model coupling.*

*See also response below.*

Add discussion and references on the topics below:

- **The Influence of Seasonal and Non-Seasonal River Release on Stratification and Sea Level Variability:** It is important to discuss how variations in river release can impact both stratification and sea level changes. Relevant studies include in addition to the already mentioned Chandanpurker et al 2022:
  - Zhang, Y. J., Ye, F., Stanev, E. V., & Grashorn, S. (2016). Seamless cross-scale modeling with SCHISM. Ocean Modelling, 102, 64-81.
  - Giffard, P., Llovel, W., Jouanno, J., Morvan, G., & Decharme, B. (2019). Contribution of the Amazon River discharge to regional sea level in the tropical Atlantic ocean. Water, 11, 2348. https://doi.org/10.3390/w11112348
  - Piecuch, C. G., Bittermann, K., Kemp, A. C., Ponte, R. M., Little, C. M., Engelhart, S. E., & Lentz, S. J. (2018). River-discharge effects on United States Atlantic and Gulf coast sea-level changes. Proceedings of the National Academy of Sciences, 115(30), 7729-7734.
  - Piecuch, C. G., & Wadehra, R. (2020). Dynamic sea level variability due to seasonal river discharge: A preliminary global ocean model study. Geophysical Research Letters, 47(4), e2020GL086984.
  - Verri, G., Pinardi, N., Oddo, P., Ciliberti, S. A., & Coppini, G. (2018). River runoff influences on the Central Mediterranean overturning circulation. Climate dynamics, 50(5-6), 1675-1703

*Response: The effect of variable river release and how it impacts coastal processes was already evoked throughout the manuscript, but we now have added more discussion in a new Section 2.1, titled "Capturing seasonal and non-seasonal river variability", as follows:*

*"Realistic (model- or observation-derived) river discharges and ancillary variables (e.g. salinity, temperature) are necessary for capturing seasonal and non-seasonal effects in the coastal ocean. The Bay of Bengal is one example where the inclusion of seasonal river discharges and salinity in regional model simulations significantly improved the representation of sea surface temperatures,*

*near-surface salinity, stratification, mixed-layer depth and barrier-layer thickness, leading to a better simulation of the formation, progression and dispersion of the freshwater plume (Jana et al., 2015).*

*Seasonal variability in river discharge not only impacts coastal salinity and temperature, but also contributes to the sea level changes both locally and remotely, mostly via a halosteric sea level contribution. This effect was observed, for example, between the mouth of the Amazon River and the continental shelves of the Gulf of Mexico and Caribbean Sea (Giffard et al., 2019). Similarly, in the U.S. Atlantic and Gulf coasts, river discharge and sea level changes were found to be significantly correlated (Piecuch et al. 2018). Such dynamic SSH signals driven by river discharge can explain 10-20% of the regional-scale seasonal variance around major rivers, such as the Amazon, Ganges, Brahmaputra, Irrawaddy, Ob, Lena, and Yenisei (Piecuch and Wadehra, 2020).*

*However, non-seasonal effects of river runoff on sea level changes remain largely unexplored over the global ocean and across a wider range of time scales, mainly due to the lack of consolidated discharge databases (Durand et al., 2019). Furthermore, river forcing must be considered jointly with wind work and heat flux, as they constitute major contributors to the energy budget in some basins (Verri et al., 2018).”*

*Further discussion on this topic was added as follows:*

- o *Section 3.1.1: “Although use of climatological data is still commonly accepted, even when the estuarine dynamics is not resolved, more realistic and less subjective estimates of volume fluxes and salinity inputs would produce a more accurate representation of coastal (e.g. river plumes) to basin-wide circulation and dynamics (e.g. dense water formation, overturning circulation cells, water exchange at straits) (Verri et al., 2018), especially during non-seasonal (e.g. storm induced) events (Chandanpurkar et al., 2022).”*
- o *Section 4: “Global systems tend to opt for climatologies in comparison with regional or coastal systems that favour observed data when available, which allows to capture the non-seasonal events and their potential local or regional impacts.”*
- o *Section 5: “Sotillo et al. (2021b) highlighted that service evolution roadmaps, such as the CMEMS guidelines, need to include a better characterization of the land boundary, especially of the coastal freshwater exchanges, to improve forecasts particularly under severe weather conditions. This includes real-time updated time series (past, present, forecasts) of river inputs for both major and minor or ephemeral streams as a progressive replacement of static climatologies.”*

- **Unstructured Modeling of the River-Sea Continuum:** This approach offers various advantages, including alleviating the challenges associated with prescribing river salinity because it can be set equal to zero at the head of an estuary solved by an unstructured grid. A dedicated section discussing this topic is warranted. Key references for this discussion include, in addition to the already mentioned Zhang et al 2016: Le Bars et al 2016, Maicu et al (2021), Bellafiore et al (2021), Vallaeys et al. (2018), Vallaeys et al. (2021), and Verri et al. (2023), Bonamano et al (2024), among many others.

*Response: We agree that this topic warrants a dedicated section; however, it was chosen to include this discussion to Section 2.3 (Freshwater input in high resolution models) instead of a new section, and to modify the subtitle as "Freshwater input in high resolution models: unstructured modelling of the river-sea continuum", since this section already had some elements suggested by the reviewers. However, this paper is not meant to provide a review on unstructured modeling, but rather on river forcing methods. Therefore, our choice is to keep this discussion as brief as possible, while highlighting the necessary key points. This is also justified by the fact that a manuscript coauthored by two of the same authors as this paper was submitted in the same collection of papers in State of the Planet. This new manuscript addresses this topic in detail and is entitled "Solving Coastal Dynamics: Introduction to High Resolution Ocean Forecasting Services", by Staneva, J., Melet, A., Veitch, J. and Matte, P.*

*We added details on unstructured modeling that relate to river forcing, in new paragraphs as follows:*

*"The use of unstructured grids offers various advantages, including a more accurate treatment of the freshwater inputs from rivers, a realistic representation of river-sea interactions and estuarine processes at spatial and temporal scales usually not resolved in the ocean, and an improved interface between estuaries and the open ocean, sometimes with higher-order spatial discretizations (Staneva et al., 2024). The unstructured grid modelling combined with an efficient vertical coordinate system can better solve the coastal sea dynamics (Verri et al., 2023).*

*"With seamless grid transitions between models or domains, flexibility and cross-scale capabilities are augmented (Zhang et al., 2016). As examples, a river-coastal-ocean continuum model has been developed for the Tiber River delta, reproducing the coastal dynamic processes better than the classic coastal–ocean representation, including the salt wedge intrusion, and revealing new features near the river mouth induced by river discharge and coastal morphology (Bonamano et al., 2024). In the Columbia River estuary, where both shelf and estuarine circulations are coupled, a multi-scale model has proved to reproduce key processes driving the river plume dynamics in a region characterized by complex bathymetry and marked gradients in density and velocity (Vallaeys et al., 2018). Likewise, Vallaeys et al. (2021) used a similar model in a topographically challenging area of the Congo River estuary, characterized by high river discharge, strong stratification and large depth. Similarly, Maicu et al. (2021) simulated the circulation in the Goro Lagoon and Po River Delta branches using downscaling and a seamless chain of models integrating local forcings and dynamics into a coarser OOFS based on a cascading approach.*

*"While these examples were successful in representing dynamical processes across temporal and spatial scales, in some contexts, the large inward tidal extent and/or complex bathymetries and coastlines, often featuring coastal infrastructures, pose significant challenges for explicitly resolving estuaries, making it impractical in many coastal models. As a result, this approach has yet to become standard practice in OOFS."*

- **Machine Learning Approaches to Estimate Riverine Release.** Key references to consider include studies that highlight successful machine learning applications in hydrology and oceanography. Regarding the salinity at river mouths some references are provided below:

- Fang, Y., wei Chen, X., Cheng, N.-S., 2017. Estuary salinity prediction using a coupled GA-SVM model: A case study of the Min River Estuary, China. Water Sci. Technol.: Water Supply 17, 52–60.
- Qiu, C., Wan, Y., 2013. Time series modeling and prediction of salinity in the Caloosahatchee River Estuary. Water Resour. Res. 49 (9), 5804–5816. http://dx.doi.org/10.1002/wrcr.20415, arXiv:https://agupubs.onlinelibrary.wiley.com/doi/pdf/10.1002/wrcr.20415, URL: https://agupubs.onlinelibrary.wiley.com/doi/ abs/10.1002/wrcr.20415.
- Guillou, N., Chapalain, G., Petton, S., 2023. Predicting sea surface salinity in a tidal estuary with machine learning. Oceanologia 65 (2), 318–332. http://dx.doi.org/10.1016/j.oceano.2022.07.007, URL: https://www.sciencedirect.com/science/article/pii/S0078323422000835.
- Qi, S., He, M., Bai, Z., Ding, Z., Sandhu, P., Zhou, Y., Namadi, P., Tom, B., Hoang, R., Anderson, J., 2022b. Multi-location emulation of a processbased salinity model using machine learning. Water 14 (13), http://dx.doi.org/10.3390/w14132030, URL: https://www.mdpi.com/2073-4441/14/13/2030
- Saccotelli, L., Verri, G., De Lorenzis, A., Cherubini, C., Caccioppoli, R., Coppini, G., Maglietta, R., 2024. Enhancing estuary salinity prediction: a Machine Learning and Deep Learning based approach. Applied Computing and Geosciences
- Maglietta, R., Verri, G., Saccotelli, L., De Lorenzis, A., Cherubini, C. Caccioppoli, R., Dimauro, G., Pinardi, N., Coppini, G. (2024) Advancing Estuarine Box Modeling: a Novel Hybrid Machine Learning and Physics-Based Approach. Environmental Modelling and Software

Regarding the water level along estuaries:
- Sampurno, J., Vallaeys, V., Ardianto, R., and Hanert, E.: Integrated hydrodynamic and machine learning models for compound flooding prediction in a data-scarce estuarine delta, Nonlin. Processes Geophys., 29, 301–315, https://doi.org/10.5194/npg-29-301-2022, 2022.

*Response: There is a vast literature on machine learning applications in hydrology and oceanography. Since our focus is on the treatment of rivers in OOFS, and indirectly on salinity, we added discussion and references, but limiting ourselves only to applications on this topic, as follows:*

- *In Section 2.2 (Freshwater input in coarse resolution models), we added the following paragraph: "New hybrid approaches, such as the Hybrid-EBM (Maglietta et al., 2025; Saccotelli et al., 2024), combine physics-based models with machine learning techniques to predict the salt-wedge intrusion length and salinity at river mouths. Hybrid-EBM outperforms the classical EBM and addresses the shortcomings of the dimensional equations in the physics-based EBM, which rely on several tunable coefficients and require site-specific calibration, by substituting them with machine learning algorithms (Maglietta et al., 2025)."*
- *In Section 2.4 (One-way and two-way coupling): "Alternative approaches for assessing the risk of compound flooding have been proposed, including integrated hydrodynamic and machine learning methods to predict water level dynamics (Sampurno et al. 2022). Such*

*approaches are particularly valuable in data-scarce regions, where developing fully calibrated, computationally intensive models can be impractical or infeasible."*

o *We added a new section 3.1.5 (Machine learning-derived discharge estimates): "Modern data-driven techniques based on machine learning are becoming increasingly used in hydrology for rainfall-runoff modelling. In particular, Long Short-Term Memory (LSTM) neural networks (Greff et al., 2016; Hochreiter and Schmidhuber, 1997) are highly effective in capturing both periodic and chaotic patterns in time-series data while accurately learning long-term dependencies (Fang et al., 2017; Hu et al., 2019; Mouatadid et al., 2019). In numerous hydrological studies, LSTM has demonstrated superior performance over traditional process-based models in simulating runoff, primarily in data-rich regions (Feng et al., 2020, 2021; Frame et al., 2022; Gauch et al., 2021; Hunt et al., 2022; Konapala et al., 2020; Kratzert et al., 2019; Lees et al., 2021; Li et al., 2023; Luppichini et al., 2024; Nearing et al., 2021; Reichstein et al., 2019). However, limited efforts have explored the transferability of LSTM models to data-scarce regions (e.g. Akpoti et al., 2024), with Ma et al., (2021) and Muhebwa et al. (2024) (and references therein) being a few such exceptions. Recent global-scale implementations (Rasiva Koya and Roy, 2024; Tang et al., 2023; Yang et al. 2023; Zhao et al. 2021) highlight the potential of LSTM models to serve as a reliable tool for global river discharge estimations. However, extensive validation outside the training basins is still required to fully assess their applicability."*

o *In Section 3.2 (Salinity and temperature): "Alternatively, salinity predictions in estuaries and at river mouths have been successfully estimated using machine learning approaches. A few examples can be found in the recent literature: Qiu and Wan (2013) developed an autoregressive model relating salinity at a given time to past observations of salinity and physical drivers (freshwater inflow, rainfall, tidal elevation) in the Caloosahatchee River Estuary; Fang et al. (2017) used a genetic algorithm coupled with support vector machine to predict salinity in the Min River Estuary; Qi et al. (2022) applied four neural network models to emulate salinity simulations in the Sacramento-San Joaquin Delta from a process-based river, estuary and land modelling system; Guillou et al. (2023) were able to reproduce the seasonal and semi-diurnal variations of sea surface salinity at the mouth of the Elorn estuary (bay of Brest), with support vector regression performing best among all tested algorithms."*

- **Estimation of River Temperature and Its Minor Role:** The estimation of river temperature should be acknowledged in the context of riverine release even if it is not the primary factor influencing oceanographic processes

*Response: In Section 3.2 (Salinity and Temperature), the following sentence was added: "Moreover, integrating salinity, temperature, and other parameters such as nutrients or sediments directly into river outflows could improve model performance (Verri et al., 2018). While these factors play a secondary role in influencing oceanographic processes, their inclusion could advance research on coastal hypoxia, carbon cycling, and regional weather and climate, ultimately supporting seamless predictions of land–ocean–atmosphere feedbacks in next-generation Earth system models (Feng et al. 2021)."*

***Minor comments:***

Line 10: I'd mention also the subsurface water discharge

*Response: We instead added in the second sentence of the abstract the following text (underlined): "They govern the hydrological and biogeochemical contributions to the coastal ocean through surface and subsurface water discharge and influence local circulation and the distribution of water masses, modulating processes such as upwelling and mixing."*

Line 40: The first time the acronym OOFS is mentioned, it should be spelled out in full.

*Response: Done.*

Line 48-49: the prescribed salinity values at river mouths are typically based on constant annual/monthly values which are the result of sensitivity tests and/or in situ campaigns (*Verri, G., Pinardi, N., Oddo, P., Ciliberti, S. A., & Coppini, G., 2018. River runoff influences on the Central Mediterranean overturning circulation. Climate dynamics, 50(5-6), 1675-1703*)

*Response: We thank the reviewer for this suggestion. This was added to the text as follows (changes are underlined): "However, although water properties at the head differ from those at the mouth, in models too coarse to resolve the estuaries, river discharge observed far from the river outlet is typically inputted at the coast with zero salinity (Verri et al., 2021; Herzfeld, 2015). Alternatively, salinity values can be prescribed based on constant annual or monthly values derived from sensitivity tests and/or in situ campaigns, when available (Verri et al., 2018)."*

Line 75: Figure 1 caption. Which EBM does the caption refer to? Please include a reference to the model

*Response: The reference was already included as a footnote. We also added this information in the caption: "Schematic diagram of the estuary box model (EBM) implemented in the Community Earth System Model (CESM) (Sun et al., 2017)."*

Line 111-112: I believe here you should refer to Verri et al 2018 rather than Verri et al 2021. Sensitivity tests by Verri et al 2018 demonstrate that a more realistic estimates of riverine inputs would produce a more accurate representation of coastal (plume) to basin wide circulation and dynamics (dense water formation, overturing circulation cells, water exchange at the straits ...)

*Response: The change was made from Verri et al. (2021) to Verri et al. (2018). We also added more details as given by the reviewer, as follows: "more realistic and less subjective estimates of volume fluxes and salinity inputs would produce a more accurate representation of coastal (e.g. river plumes) to basin-wide circulation and dynamics (e.g. dense water formation, overturning circulation cells, water exchange at straits) (Verri et al., 2018)"*

Line 192: "However, strong land-sea differences in microwave emissivity make satellite observations unreliable within some 70 km of the coast" I believe the water turbidity is the main limit to be mentioned here

*Response: The article by Vazquez-Cuervo et al. (2018) describes land contamination as the primary factor for degraded quality in the satellite salinity products near the coast. In addition, Menezes (2020) showed*

*that there can be seasonal sensitivity in SMAP's skill, notably in the Bay of Bengal. This was also stressed by Grodsky et al. (2018) in an application in the Gulf of Maine, showing that SSS retrievals over cold coastal seas are subject to an SST-dependent bias due to microwave sensor sensitivity, on top of a land contamination bias.*

*The sentence was modified as follows (changes are underlined): "However, seasonal variability in the skill of SSS retrievals can be associated with SST-dependent bias and strong land-sea differences in microwave emissivity, making satellite observations unreliable within some 70 km of the coast (Grodsky et al., 2018; Menezes, 2020; Vazquez-Cuervo et al., 2018)."*

*Menezes, V. v. (2020). Statistical assessment of sea-surface salinity from SMAP: Arabian sea, bay of Bengal and a promising red sea application. Remote Sensing, 12(3). https://doi.org/10.3390/rs12030447*

*Grodsky, S. A., Vandemark, D., & Feng, H. (2018). Assessing coastal SMAP surface salinity accuracy and its application to monitoring Gulf of Maine circulation dynamics. Remote Sensing, 10(8). https://doi.org/10.3390/rs10081232*

Line 194-195: The satellite retrieved salinity close to the river mouth is a crucial challenge and more recent studies should be mentioned and briefly discussed *e.g. Medina et al 2020; Sakai et al 2021, Chen et al 2017*

*Response: We thank the reviewer for the suggestions. The following additions were made to the text and references: "Higher resolution coastal satellite products have been demonstrated based on empirical relationships between local salinity and ocean color observations (Geiger et al, 2011; Chen et al., 2017), using deep neural networks trained on Sentinel-2 Level 1-C Top of Atmosphere (TOA) reflectance data (Medina-Lopez and Ureña-Fuentes, 2019; Medina-Lopez, 2020), or by relating the reflectance of the visible bands from Sentinel-2 imagery with electrical conductivity, influenced by the concentration and composition of dissolved salts (Sakai et al., 2021), although these are not applied globally"*

*Furthermore, we added the following paragraph: "A recent study in the German Bight (Thao et al., 2024) demonstrated the critical role of high-resolution salinity inputs at estuarine mouths in improving the predictive capabilities of coupled wave-ocean models. Using the GCOAST model system, which seamlessly integrates estuarine and coastal dynamics with regional ocean models, researchers validated salinity and temperature fields against in-situ observations. The results highlighted that estuarine inflows significantly enhance the accuracy of coastal ocean models."*

*Chen, S., & Hu, C. Estimating sea surface salinity in the northern Gulf of Mexico from satellite ocean color measurements. Remote Sensing of Environment, 201, 115–132. https://doi.org/https://doi.org/10.1016/j.rse.2017.09.004, 2017.*

*Medina-Lopez, E. Machine learning and the end of atmospheric corrections: A comparison between high-resolution sea surface salinity in coastal areas from top and bottom of atmosphere Sentinel-2 imagery. Remote Sensing, 12(18). https://doi.org/10.3390/RS12182924, 2020.*

*Medina-Lopez, E., & Ureña-Fuentes, L. High-resolution sea surface temperature and salinity in coastal areas worldwide from raw satellite data. Remote Sensing, 11(19). https://doi.org/10.3390/rs11192191, 2019.*

*Sakai, T., Omori, K., Oo, A. N., & Zaw, Y. N. Monitoring saline intrusion in the Ayeyarwady Delta, Myanmar, using data from the Sentinel-2 satellite mission. Paddy and Water Environment, 19(2), 283–294. https://doi.org/10.1007/s10333-020-00837-0, 2021.*

*Thao, N.T., Staneva, J., Grayek, S., Bonaduce, A., Hagemann, S., Nam, T.P., Kumar, R., & Rakovec, O. (2024): Impacts of extreme river discharge on coastal dynamics and environment: Insights from high-resolution modeling in the German Bight. Regional Studies in Marine Science, Vol 73, 103476, doi:10.1016/j.rsma.2024.103476*

Section 3.1.2: a missing reference in the list is the recent database of climatological runoff for the Adriatic rivers provided by *Aragão, L., Mentaschi, L., Pinardi, N., Verri, G., Senatore, A., & Di Sabatino, S. (2024). The freshwater discharge into the Adriatic Sea revisited. Frontiers in Climate*

*Response: We thank the reviewer for this suggestion. The following text was added to the list of regional databases in Section 3.1.2:*

*• A river discharge climatology and corresponding historical time series for all rivers flowing into the Adriatic Sea with an average climatological daily discharge exceeding 1 m3s−1 (Aragão et al., 2024).*

*As well as this reference:*

*• Long-term (1993-2011) satellite-derived estimates of continental freshwater discharge into the Bay of Bengal (Papa et al., 2012).*

Section 3.3 all the OOFS should be references through links to their web pages /publications

*Response: Links to websites or publications were added for each OOFS.*

---

## Author Comment (AC3)

**RESPONSE TO REVIEWER #3'S COMMENTS**

**General Comments:**

This paper provides a useful review of methods and data available for providing river runoff forcing into operational ocean forecast models (OOFS). The text is clear and accessible to a wide audience. While not an exhaustive list, the summary tables also provide useful reference of different operational configurations. However, my primary concern is that the review is missing a discussion and conclusion. It would greatly benefit from comparison of the different approaches presented, discussion on challenges for operational implementation, and summarising priorities for future work e.g., considering recent R&D that could be brought through to future OOFS.

Further comments relating to specific aspects of the text are listed below.

*Response: We thank the Reviewer for this positive and constructive feedback on our manuscript. In response to these general comments, we made the following additions to the manuscript:*

- *We added in Section 4 (formerly Section 3.3) a synthesis from the survey of the different methods and data sources used in current OOFS as a means for comparison of the different approaches presented.*
- *We added a discussion and conclusion in Section 5 (Summary and recommendations). In this section, we highlight some of the major limitations and challenges in representing rivers in current OOFS. We also provide recommendations for future research and developments.*

*However, our objective in this paper is to make an inventory of existing current approaches and available datasets, rather than providing a detailed gap analysis with recommendations on ways forward, which will be the topic of a follow-up contribution.*

**Specific Comments:**

Abstract: Clarify that here you focus on physical river forcing rather than supply of nutrients or other materials.

*Response: In the abstract, we modified the following sentence: "This paper provides an overview of recent  approaches to representing coastal river discharges and processes in ocean models, with a particular focus on estuaries." We also modified the following sentence: "A review of river data availability is also presented, illustrating various sources of freshwater discharge and salinity […]".*

Line 17: I would not refer to the tables as a "compendium" as this suggests a complete summary. I would instead simply say that you present responses from a survey of existing OOFS providers.

*Response: The sentence was reformulated as follows: "In addition, responses from a survey of existing operational ocean forecasting systems (OOFS) providers are synthetized, with a focus on how river forcing is treated, from global to coastal scales."*

Line 26-27: This suggests upwelling is the only way that productivity is impacted, whereas all the above may impact productivity. Suggest rephrasing.

*Response: The sentence was modified as follows: "Freshwater inputs to the ocean also modulate coastal upwelling events. Altogether,  these factors impact productivity of the coastal marine environment (Sotillo et al., 2021a)."*

Also, please clarify that input of nutrients and other parameters that would impact productivity are neglected within this review, but are clearly important for consideration in future work.

*Response: We added the following sentence in the introduction, before introducing the different sections: "The objective of this paper is to provide an inventory of existing methods and available datasets adopted in current operational ocean forecasting systems (OOFS) for representing river forcing. As the focus is on freshwater discharges, the river supply of nutrients and other materials are neglected in this review, but are partly addressed in a separate contribution by Cossarini et al. (2024)."*

*In the conclusions, we added recommendations on having "standardized inputs of freshwater (and associated river inputs of nutrients and sediment loading)".*

*Cossarini, G., Moore, A., Ciavatta, S., & Fennel, K. (2024). Numerical Models for Monitoring and Forecasting Ocean Biogeochemistry: a short description of present status. State Planet Discuss., 2024, 1–13. https://doi.org/10.5194/sp-2024-8*

Line 33: The use of climatology may not always be limited to coarse resolution models (as suggested here)?

*Response: We agree, although this was not exactly the intended meaning. For clarity, the sentence was modified as follows: "At coarse scales that cannot resolve the estuarine dynamics, but even at finer scales in some cases, river outlets are  often represented in a simplistic way, with climatological runoff and zero or constant salinity values, implicitly neglecting estuarine mixing or exchange"*

Line 87-88: Even where resolution may be higher than estuary width, explicitly resolving the estuary may still be challenging and therefore unlikely in many coastal models, e.g., due to large inward tidal extent and/or complex coastlines or coastal infrastructure.

*Response: True. We added the following sentence: "[...] the large inward tidal extent and/or complex bathymetries and coastlines, often featuring coastal infrastructures, pose significant challenges for explicitly resolving estuaries, making it impractical in many coastal models."*

Section 2.3: This is an active area of research, so worth clarifying where references refer to use in operational systems vs. ongoing R&D configurations? Further discussion on scope for future development here could be useful in an added summary/discussion section (e.g., is computational cost the main barrier for this approach?).

*Response: We agree; most examples correspond to R&D configurations. The following sentence was added in Section 2.2: "however, this approach has yet to become standard practice in OOFS".*

*We also added a summary/recommendation section where recommendations include the need for "standardized inputs of freshwater (and associated river inputs of nutrients and sediment loading), homogenized river forcing approaches, and a more integrated watershed-ocean strategy".*

Line 114: Worth acknowledging here that for some countries even larger rivers may lack routine monitoring (for both historical and NRT data).

*Response: We made the following addition (underlined): "Moreover, given the global decline of the hydrometric networks, building climatologies is not always possible, especially for small or less- studied rivers, and even for large rivers in regions where routine monitoring is absent."*

Section 3.1.2: While the list of possible datasets is useful, a discussion on the various options and pros/cons for different approaches could be useful? For example, could you present similar methods together (e.g., in situ vs remote sensing), and discuss which are static vs. updated?

*Response: Our objective in this paper is not to assess the quality or value of a given product or dataset, but rather to provide an inventory of existing methods and available datasets adopted in current OOFS for representing river forcing. This is now made clearer in the introduction.*

*In Section 3.1.2, we regrouped the list of datasets into sub-groups as follows: in situ databases (4), model-derived databases (3), hybrid database (1), and satellite-based database (1).*

*Finally, in Section 3.1.2, we had already provided some elements of discussion, repeated here (additions are underlined):*

- *"Of particular importance is the fact that some of these databases use model-simulated runoff ratios (e.g. from Community Land Model (CLM) or river routing model) over gauged and ungauged drainage areas to estimate the contribution from the areas not monitored by the hydrometric network and adjust the station flow to represent river mouth outflow (e.g. Dai et al. (2009). This allows more precise derivation of the total discharge into the global oceans, through the sum of both gauged and ungauged discharges."*
- *"Unless explicitly stated (e.g. for EMODnet Physics), it is not evident that any of these databases are updated on a regular schedule; some remain static, others are updated irregularly (at an unknown frequency). Such databases are useful in the context of a reanalysis, but less so in an operational context where near-real-time data feeds are required. Furthermore, a detailed comparative assessment of these various data sources is still lacking."*

Line 137-138: I think these two sources should be referred to separately as provide regional rather than global datasets.

*Response: Agreed. We separated these two sources and added two new sources, as follows:*

*"Regional databases also exist, such as:*

- *(NEW) Long-term (1993-2011) satellite-derived estimates of continental freshwater discharge into the Bay of Bengal (Papa et al., 2012).*
- *A database of pan-Arctic river discharge (R-Arcticnet: https://www.r-arcticnet.sr.unh.edu/v4.0/index.html).*
- *A database for Greenland liquid water discharge from 1958 through 2019 (Mankoff et al., 2020).*
- *(NEW) A river discharge climatology and corresponding historical time series for all rivers flowing into the Adriatic Sea with an average climatological daily discharge exceeding 1 m3s−1 (Aragão et al., 2024).*

Line 170-172: How many of the other products are actually freely available? (For both historical and NRT?)

*Response: We have not documented the type of licence that comes with the data, or whether these are freely available, for each of the listed operational products. No actions taken.*

Section 3.2: I expected this section to have more discussion around the forcing of T/S at river outflow, rather than purely model validation? While it's worth stressing that tuning models based on incorrect data is an issue, it's also worth reiterating that having T/S data within river outflows (e.g., from hydrological runoff models?) could help avoid this issue. The same would apply for outflow of nutrients and other parameters of course.

*Response: We agree. We added the following in Section 3.2:*

*"A recent study in the German Bight (Thao et al., 2024) demonstrated the critical role of high-resolution salinity inputs at estuarine mouths in improving the predictive capabilities of coupled wave-ocean models. Using the GCOAST model system, which seamlessly integrates estuarine and coastal dynamics with regional ocean models, researchers validated salinity and temperature fields against in-situ observations. The results highlighted that estuarine inflows significantly enhance the accuracy of coastal ocean models."*

*[...]*

*"Moreover, integrating salinity, temperature, and other parameters such as nutrients or sediments directly into river outflows could improve model performance (Verri et al., 2018; Thao et al., 2024). While these factors play a secondary role in influencing oceanographic processes, their inclusion could advance research on coastal hypoxia, carbon cycling, and regional weather and climate, ultimately supporting seamless predictions of land–ocean–atmosphere feedbacks in next-generation Earth system models (Feng et al. 2021)."*

*Also, in Section 4 we added a comment from the survey responses regarding T/S, as follows: "Salinity and temperature of the input freshwater can either be set to zero and to the local SST, respectively, or derived from a combination of real-time gauge data and monthly averages when available."*

Section 3.3: This currently feels like an odd way to complete the review. There is a need to provide a summary and discussion section. For example, summarising the current "state of the art" developments, limitations of existing OOFS, and priorities for future work? To help this flow, I feel the table provided should be included as an appendix (and ideally in landscape format, to assist reading columns with more content).

*Response: We significantly expanded Section 4 (formerly Section 3.3) by providing a summary of the current OOFS methods and data sources from the survey, both in the text and with a new figure. Also, we added a Section 5 (Summary and recommendations) that briefly addresses the limitations of existing OOFS and priorities for future work (see response to Reviewer 1). The tables were also moved to a new Appendix A in landscape format (in the original version, landscape format was not allowed at the submission stage).*

Within each of the tables, for each of the systems it would be useful to understand whether there are references to the data source and/or publications? If this isn't available from the survey, then noting simply the responding institution or provider for each OOFS would be useful to provide a contact for

further information. Please also clarify whether responses to the survey have been summarised/rephrased or provided as given (I presume the latter, but need to clarify).

*Response: We tried to find official references for each system, but the information is often hidden in multiple papers and not always easy to find. Instead, we added one column for the institution/provider. Links to the systems webpages, or relevant data source, are included as footnotes for each system.*

*The responses to the survey are reported as given by the participants; nearly no changes were made to each contributed entry, except for a few added references and acronym definitions. This is now explained in the introduction to Appendix A, as follows:*

*"This Appendix presents results of a survey conducted among members of the OceanPredict community in May 2023. The responses are reported in the following tables as given by the participants; nearly no changes were made to each contributed entry, except for a few added references and acronym definitions."*

Section 3.3.3: How did you differentiate between coastal vs regional domains?

*Response: Coastal or regional domains were differentiated by respondents who filled in the tables.*

**Technical Corrections:**

Line 40: OOFS acronym introduced without being defined.

*Response: Corrected. The OOFS acronym is now introduced.*

Line 111: Suggest rephrasing "freshwater and salinity" to "volume fluxes and salinity" inputs?

*Response: Corrected as suggested.*

Figure 2: Please clarify whether river networks shown come from GloFAS?

*Response: "River networks come from GloFAS" was added to the figure caption.*

Hyperlinks: Throughout the text there appear to be hyperlinks to relevant data sources. However, these links don't work. Please ensure they do in the revised version, or make sure to reference in an alternative format.

*Response: Thank you for this comment. Apparently, the links were lost during conversion to PDF. Full web addresses are now added in the text as footnotes.*

Acronyms: There are multiple within the survey responses. Please define where possible.

*Response: All acronyms were defined where possible.*

---

## Author Response (AR1)

Dear Editor,

A point-by-point response to each of the reviewers' comments has been provided in the discussion. The reported modifications have been integrated in the revised manuscript. Please refer to the track change document for detailed corrections.

Best regards,

Pascal Matte, on behalf of coauthors.

---

## Author Response (AR2)

Dear Editor,

In the section "Authors contribution" we now use initials for the authors' names. No further corrections were made. Thank you for your comments.

Best regards,

Pascal Matte, on behalf of coauthors.